# Complete suspension culture of human induced pluripotent stem cells supplemented with suppressors of spontaneous differentiation

Mami Matsuo-Takasaki[1], Sho Kambayashi[2], Yasuko Hemmi[1], Tamami Wakabayashi[1], Tomoya Shimizu[1], Yuri An[1], Hidenori Ito[1], Kazuhiro Takeuchi[2], Masato Ibuki[2], Terasu Kawashima[2], Rio Masayasu[2], Manami Suzuki[2], Yoshikazu Kawai[2], Masafumi Umekage[3], Tomoaki M Kato[3], Michiya Noguchi[4], Koji Nakade[5†], Yukio Nakamura[4], Tomoyuki Nakaishi[2], Naoki Nishishita[2], Masayoshi Tsukahara[3], Yohei Hayashi[1,6]*

[1]iPS Cell Advanced Characterization and Development Team, RIKEN BioResource Research Center, Ibaraki, Japan; [2]Regenerative Medicine and Cell Therapy Laboratories, KANEKA CORPORATION, Kobe, Japan; [3]Research and Development Center, CiRA Foundation, Kyoto, Japan; [4]Cell Engineering Division, RIKEN BioResource Research Center, Ibaraki, Japan; [5]Gene Engineering Division, RIKEN BioResource Research Center, Ibaraki, Japan; [6]Faculty of Medicine and School of Integrative and Global Majors, University of Tsukuba, Ibaraki, Japan

*For correspondence: yohei.hayashi@riken.jp

Present address: †Tokyo Cell Medical Science Institute, Tokyo, Japan

## eLife Assessment

This comprehensive and **compelling** study presents a robust, cost-effective method for expanding pluripotent stem cells. The authors have identified a media condition that maintains iPSCs in suspension cultures by inhibiting the PKCβ and Wnt signaling pathways. The article is **important** for the pluripotent stem cell field as it seeks robust and economical approaches to expand iPSCs at scale for high-throughput screens and preclinical studies. While the authors have tested their media and protocol on a few lines, given the variability of iPSCs, further testing across more cell lines and in different laboratory settings will be crucial to evaluate its reproducibility.

**Abstract** Human induced pluripotent stem cells (hiPSCs) are promising resources for producing various types of tissues in regenerative medicine; however, the improvement in a scalable culture system that can precisely control the cellular status of hiPSCs is needed. Utilizing suspension culture without microcarriers or special materials allows for massive production, automation, cost-effectiveness, and safety assurance in industrialized regenerative medicine. Here, we found that hiPSCs cultured in suspension conditions with continuous agitation without microcarriers or extracellular matrix components were more prone to spontaneous differentiation than those cultured in conventional adherent conditions. Adding PKCβ and Wnt signaling pathway inhibitors in the suspension conditions suppressed the spontaneous differentiation of hiPSCs into ectoderm and mesendoderm, respectively. In these conditions, we successfully completed the culture processes of hiPSCs, including the generation of hiPSCs from peripheral blood mononuclear cells with the expansion of bulk population and single-cell sorted clones, long-term culture with robust self-renewal characteristics, single-cell cloning, direct cryopreservation from suspension culture and their successful recovery, and efficient mass production of a clinical-grade hiPSC line. Our results demonstrate that

precise control of the cellular status in suspension culture conditions paves the way for their stable and automated clinical application.

## Introduction

Human induced pluripotent stem cells (hiPSCs) are promising resources for various types of tissues in regenerative medicine (*Takahashi et al., 2007*; *Yu et al., 2007*). To enable cell therapy from hiPSCs, the development of a large-scale manufacturing system is essential because massive cell numbers are required to compose transplantable cells which are enough to rescue the desired physiological function (*Chen et al., 2014*; *Kim and Kino-Oka, 2020*; *Tannenbaum and Reubinoff, 2022*). In general, hiPSCs are believed to possess their scaffold dependency and are cultured under adhesion and monolayer culture conditions (*Hayashi and Furue, 2016*; *Xu et al., 2001*). However, utilizing suspension culture without microcarriers or special materials allows for massive production, automation, cost-effectiveness, and safety assurance in industrialized regenerative medicine.

Several attempts have been made to develop suspension culture technologies enabling rapid and large-scale preparation of hiPSCs (*Amit et al., 2010*; *Amit et al., 2011*; *Dang et al., 2004*; *Elanzew*

**Table 1.** The list of published studies on scalable suspension culture without microcarriers for human pluripotent stem cells (hPSCs).

| Article | Medium | Additives | Cell lines tested | Scalability | Transcriptome | Single-cell cloning | Direct freeze and thaw | iPSC generation |
|---|---|---|---|---|---|---|---|---|
| *Amit et al., 2010*; *Amit et al., 2011* | DMEM/F12 + KSR | Y-27632, LIF, bFGF | hESC (I3, Ie, I5, H9.2, H9, H7, H14) hiPSC (iF4, J1.2.3, C3, C2, KTN7, KTN13) | $\sim 10^7$ | No | No | No | No |
| *Krawetz et al., 2010* | mTeSR1 | Y-27632, Rapamycin | hESC (H9) | $\sim 10^8$ | No | No | No | No |
| *Singh et al., 2010*; *Zweigerdt et al., 2011*; *Olmer et al., 2012* | KnockOut DMEM + KSR, mTeSR1 | Y-27632, bFGF | hESC (hES2, hES3, and ESI049) hiPSC (hCBiPS2, hiPSOCT4eGFP) | $\sim 10^8$ | No | No | No | No |
| *Steiner et al., 2010* | Neurobasal medium + KSR | Y-27632, bFGF, Activin A, Fibronectin, Gelatin, BDNF, NT3, NT4, Nutridoma-CS | hESC (HES1, HES2, H7) | $\sim 10^6$ | No | No | No | No (of note, three hESC lines derivation) |
| *Wang et al., 2013* | TeSR-E8 | Y-27632 | hiPSC (BC1, TNC1) | $\sim 10^8$ | No | No | No | No |
| *Hunt et al., 2014* | mTeSR1 | Y-27632 | hESC (H1, H9) | $\sim 10^7$ | No | No | No | No |
| *Elanzew et al., 2015* | mTeSR1, TeSR-E8 | Y-27632 | hESC (H9) and hiPSC (iLB-C-31f-r1) | $\sim 10^7$ | No | No | No | No |
| *Horiguchi and Sakai, 2016*; *Ibuki et al., 2019* | TeSR-E8, mTeSR1 | Y-27632, KSR, Albumax, LPA, S1P, | hiPSC (TkDN4-M, TkDA3-4, 201B7) | $\sim 10^6$ | No | No | No | No |
| *Nath et al., 2017* | Dialyzed DMEM/F-12 | Y-27632, bFGF, TGF-b1 | hiPSC (Tic) | $\sim 10^8$ | No | No | No | No |
| *Kwok et al., 2018* | mTeSR1, StemMACS | Y-27632 | hiPSC (AR1034ZIMA hiPSC clone1, FS hiPSC clone2) | $\sim 10^9$ | No | No | No | No |
| *Lipsitz et al., 2018* | Nutristem, DMEM/F12+KSR | Y-27632, LIF, TGFβ1 FGF2, CHIR99021 SP600125, BIRB796, Gö6983 | hESC (H9, HES2, WIB) and hiPSC (C1.15) | $\sim 10^7$ | No | No | No | No |
| *Rohani et al., 2020* | mTeSR1, RSeT | Y-27632, Rapamycin | hESC (H1, H9) | $\sim 10^7$ | Yes | No | No | No |
| This study | AK02N, AK03N, StemScale, mTeSR1 | Y-27632, IWR-1-endo, LY333531 | hiPSC (WTC11, 201B7, 1383D6, 1231A3, HiPS-NB1RGB, Ff-I14s04) | $\sim 10^9$ (in 3 passages) | Yes | Yes | Yes | Yes |

*et al., 2015; Horiguchi and Sakai, 2016; Hunt et al., 2014; Ibuki et al., 2019; Kehoe et al., 2010; Kim et al., 2019; Krawetz et al., 2010; Kwok et al., 2018; Lam et al., 2016; Lipsitz et al., 2018; Nath et al., 2017; Oh et al., 2009; Olmer et al., 2012; Rohani et al., 2020; Shafa et al., 2012; Singh et al., 2010; Steiner et al., 2010; Wang et al., 2013; Zweigerdt et al., 2011;* summarized in *Table 1*). These studies have achieved long-term culture and/or mass expansion of hiPSCs and/or human embryonic stem cells (hESCs) in suspension conditions. However, completed processes from clonal hiPSC generation to mass production of hiPSCs based on the precise control of cell status have not yet been achieved.

In this study, we have investigated what hampers the stable maintenance of undifferentiated cell states in suspension conditions. hiPSCs cultured in suspension conditions with continuous agitation without any microcarriers or extracellular matrix (ECM) components were more prone to spontaneous differentiation than those cultured in conventional adherent conditions. From screening of candidate molecules to suppress the spontaneous differentiation of hiPSCs, we have identified that inhibitors of PKCβ and Wnt signaling pathways suppress their differentiation into ectoderm and mesendoderm, respectively. In these conditions, we aimed to complete the processes of handling hiPSCs including the generation of hiPSCs with the expansion of bulk population and single-cell sorted clones, long-term culture with robust self-renewal characteristics, single-cell cloning, direct cryopreservation from suspension conditions and their successful recovery, and efficient mass production of a clinical-grade hiPSC line.

## Results

### Suspension-cultured hiPSCs are prone to spontaneous differentiation

First, we investigated whether the quality of hiPSCs in suspension and adherent conditions is equivalent or not. hiPSCs (WTC11 line) were cultured in a conventional medium, StemFit AK02N (Ajinomoto, Tokyo, Japan), with continuous agitation (90 rpm) in non-adhesive cell culture plates for two passages (5 days during passages) and examined (*Figure 1A*). In suspension conditions on days 5 and 10, hiPSCs formed round cell assemblies with slightly uneven surfaces (*Figure 1B*). Gene expression analysis with RT-qPCR revealed that the expression of differentiation markers, such as *PAX6* (ectoderm), *SOX17* (endoderm), and *T* (mesoderm), increased in suspension-cultured hiPSCs for 10 days (*Figure 1C*). To monitor the spontaneous differentiation at single-cell resolution, we established knock-in reporter hiPSC lines of PAX6-tdTomato and SOX17-tdTomato to visualize and quantify the expression of PAX6 and SOX17 at the protein level, respectively (*Figure 1—figure supplement 1A–H*). tdTomato-positive cells were clearly observed in day 10 samples in suspension-cultured hiPSCs, whereas no fluorescent-positive cells were observed in adherent culture conditions (*Figure 1D*). Flow cytometric analysis revealed that hiPSCs in the suspension conditions contained non-negligible percentages of PAX6-tdTomato-positive and SOX17-tdTomato-positive cells (*Figure 1E and F*). Western blot analysis revealed that protein expression of PAX6 and SOX17 in the suspension conditions was significantly increased (*Figure 2—figure supplement 1A–D*). The ratio of positive cells for a cell surface marker for undifferentiated hiPSCs, TRA1-60, was significantly lower in the suspension conditions (*Figure 2—figure supplement 1E and F*). These results suggest that a portion of hiPSCs are spontaneously differentiated in the suspension conditions when cultured in conventional media. To examine global changes in gene expression patterns between suspension and adherent conditions, whole-transcriptomic RNA-seq experiments with statistical tests were performed. Gene Set Enrichment Analysis (GSEA) of all the genes and Gene Ontology Enrichment Analysis (GOEA) on differentially regulated genes revealed that, in the suspension conditions, many genes involved in differentiation toward various tissues and cell–cell adhesions were significantly upregulated. In contrast, genes involved in nucleotide metabolism, hypoxic responses, and ECM organization were downregulated significantly (*Figure 1G–J*). These results suggest that hiPSCs in the suspension conditions are in the process of spontaneous differentiation into various cell lineages and are characterized by specific signatures of gene expression patterns.

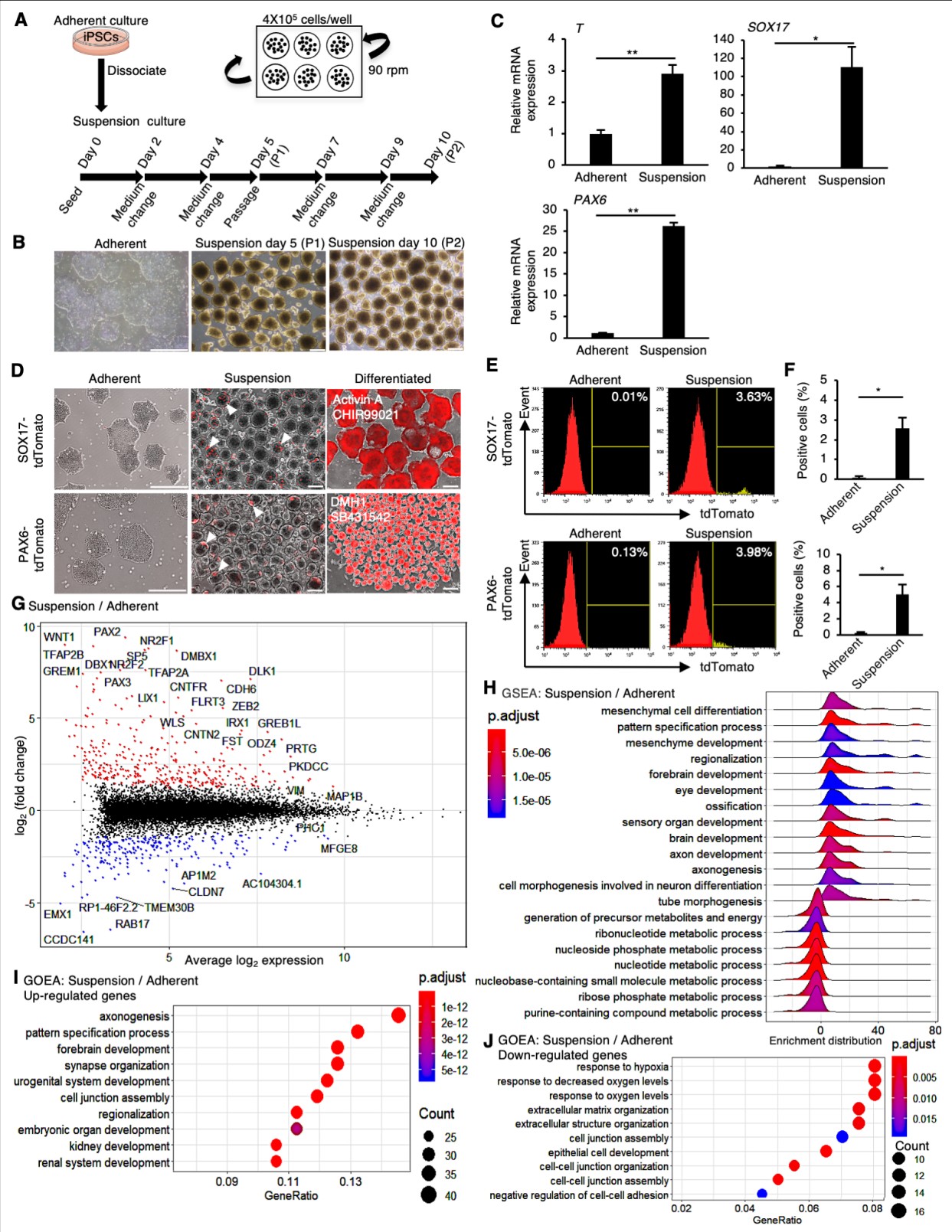

**Figure 1.** Human induced pluripotent stem cells (hiPSCs) maintained under suspension conditions undergo spontaneous differentiation. (**A**) Schematics representing hiPSCs in suspension conditions. (**B**) Phase-contrast images of adherent- or suspension-cultured hiPSCs on day 5 (passage 1 [P1]) and day 10 (passage 2 [P2]). Scale bars: 400 μm. (**C**) Gene expression in hiPSCs cultured under adherent or suspension conditions on P2. Bar graphs show the mean ± SE (n = 3). p-values were statistically analyzed with Student's t-test. (**D**) Phase-contrast and fluorescent images of adherent or suspension-

*Figure 1 continued on next page*

*Figure 1 continued*

cultured reporter hiPSCs on P2. White arrowheads indicate spontaneous expression of SOX17 and PAX6 in suspension conditions. Scale bars: 400 μm. (**E**) Quantification of hiPSCs spontaneous differentiation with flow cytometry. (**F**) Averaged tdTomato-positive cell ratio (%) from flow cytometry data (mean ± SE from n = 3). p-values were statistically analyzed with Student's *t*-test. (**G**) An MA plot (log2 fold change versus mean average expression) comparing transcriptomes between suspension and adherent conditions from RNA-seq data. The representative gene name is shown in the plot. (**H**) Gene Set Enrichment Analysis (GSEA) on the gene sets of suspension-cultured hiPSCs to adherent cultures. Adjusted p-values are shown as blue to red from low to high values. (**I, J**) Gene Ontology Enrichment Analysis (GOEA) on the gene sets of suspension-cultured hiPSCs to adherent culture. Results are ranked by significance (p-adjusted value) and/or counted gene numbers. * or ** in the graphs indicate p<0.05 or p<0.01, respectively.

The online version of this article includes the following source data and figure supplement(s) for figure 1:

**Figure supplement 1.** Characterization of PAX6-tdTomato and SOX17-tdTomato reporter lines.

**Figure supplement 1—source data 1.** File containing original gel images of agarose gel electrophoresis for *Figure 1—figure supplement 1B*, indicating the relevant bands from genotyping PCR bands.

**Figure supplement 1—source data 2.** Original files gel images of agarose gel electrophoresis for *Figure 1—figure supplement 1B*.

## Wnt signaling inhibitors suppress spontaneous mesendodermal differentiation in suspension conditions

We next aimed to identify the factors that inhibit the spontaneous differentiation of hiPSCs in suspension conditions (*Figure 2A*). As a candidate for the inducer of spontaneous differentiation, Wnt signaling induces the differentiation of human pluripotent stem cells into mesendoderm lineages exogenously (*Nakanishi et al., 2009*; *Sumi et al., 2008*; *Tran et al., 2009*; *Vijayaragavan et al., 2009*; *Woll et al., 2008*). Also, endogenous expression and activation of Wnt signaling in pluripotent stem cells are involved in the regulation of mesendoderm differentiation potentials (*Dziedzicka et al., 2021*). Thus, we hypothesized that adding Wnt signaling inhibitors/activators may alter the spontaneous differentiation of hiPSCs into mesendoderm. Therefore, Wnt signaling inhibitors, IWP2 or IWR-1-endo (*Chen et al., 2009*), or an activator, CHIR99021 (*Ring et al., 2003*), were added to the culture medium under suspension conditions. hiPSC aggregates treated with or without Wnt inhibitors showed similar round shapes (*Figure 2B*). In contrast, hiPSC aggregates treated with CHIR99021 formed heterogeneously shaped cyst-like structures, suggesting that these cells were largely differentiated. In samples treated with inhibitors, both T and SOX17 expression levels were significantly reduced to the level of adherent-cultured hiPSCs; however, there was only a small reduction in *PAX6* expression in the IWR-1-endo-treated condition and no reduction in the IWP2-treated condition (*Figure 2C*). Conversely, CHIR99021-treated cell aggregates showed markedly increased T and *SOX17* expression and decreased *OCT4* expression. Additionally, SOX17 protein expression was suppressed in hiPSCs treated with IWR-1-endo in suspension conditions, although its expression increased in hiPSCs in suspension conditions with conventional culture medium compared to adherent conditions (*Figure 2—figure supplement 1A and B*). These results indicate that Wnt signaling inhibitors effectively suppress mesendodermal differentiation in suspension conditions, but are insufficient to suppress ectodermal differentiation.

## PKC signal inhibitors suppress spontaneous neuroectodermal differentiation in suspension conditions

To identify molecules with inhibitory activity on neuroectodermal differentiation, hiPSCs were treated with candidate molecules in suspension conditions. We selected these candidate molecules based on previous studies related to signaling pathways or epigenetic regulations in neuroectodermal development (reviewed in *Giacoman-Lozano et al., 2022*; *Imaizumi and Okano, 2021*; *Sasai et al., 2021*; *Stern, 2024*) or in pluripotency safeguards (reviewed in *Hackett and Surani, 2014*; *Li and Belmonte, 2017*; *Takahashi and Yamanaka, 2016*; *Yagi et al., 2017*; *Figure 2A*; listed in *Supplementary file 1*). Out of the candidate molecules tested, Gö6983, a pan-PKC inhibitor (*Gschwendt et al., 1996*), and BMP4 showed strong inhibition effects on *PAX6* expression (*Figure 2D*). Further, simultaneous treatment with Gö6983 and IWR-1-endo decreased *PAX6*, *T*, and *SOX17* expression, while maintaining *OCT4* expression. To confirm these screening test results, the dose-dependent effect of Gö6983 or another PKCα, β, γ inhibitor GF109203X (GFX) (*Toullec et al., 1991*) on the inhibition of *PAX6* expression was observed at constant concentrations of IWR-1-endo (*Figure 2E and F*). These results demonstrate that the inhibition of PKC signaling pathway effectively suppresses

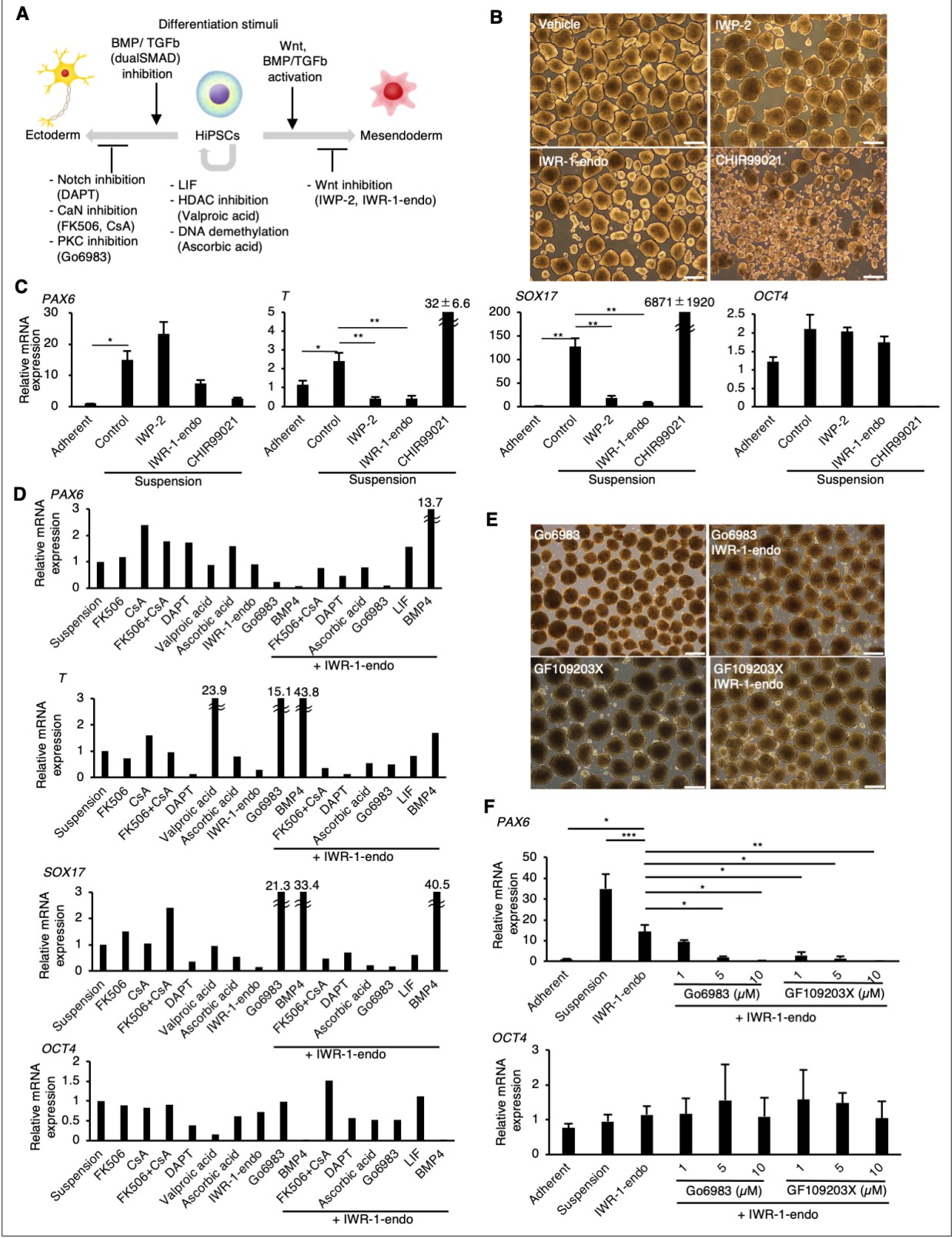

**Figure 2.** PKC inhibitors suppress spontaneous differentiation of human induced pluripotent stem cells (hiPSCs) into neural ectoderm in suspension conditions. (**A**) Schematics of the factors related to the self-renewal and early differentiation of hiPSCs. (**B**) Phase-contrast pictures of suspension-cultured hiPSCs in the presence of Wnt signaling inhibitors (IWP-2 and IWR-1-endo) or activator (CHIR99021). Scale bars: 400 µm. (**C**) Gene expression of hiPSCs in suspension conditions with or without IWP2, IWR-1-endo, or CHIR99021. RT-qPCR was performed on day 10 samples (P2). Gene expressions

*Figure 2 continued on next page*

*Figure 2 continued*

were normalized to GAPDH and displayed as relative fold increase to adherent-cultured samples. Bar graphs show the mean ± SE. p-values were statistically analyzed using Dunnett's multiple comparisons test. (**D**) Screening of inhibitory activity of candidate molecules on neuroectoderm differentiation in suspension-cultured hiPSCs. Candidate molecules were added in combination as shown. Results are displayed as relative fold increase to suspension-cultured samples without pharmacological treatment. n = 1. (**E**) Phase-contrast images of suspension-cultured hiPSCs on P2 in the presence of PKC inhibitors (Gö6983 or GF109203X) alone, or in combination with IWR-1-endo. Scale bars: 400μm. (**F**) The gene expression in suspension-cultured hiPSCs in the presence of IWR-1-endo or with combined IWR-1-endo and different doses of PKC inhibitors. Results are displayed as relative fold increase to adherent-culture. Data are presented as mean ± SE (n = 3). *, **, or *** in the graphs indicate p<0.05, p<0.01, or p<0.001, respectively.

The online version of this article includes the following source data and figure supplement(s) for figure 2:

**Figure supplement 1.** Effects of PKCβ and IWR-1-endo on suspension-cultured human induced pluripotent stem cells (hiPSCs) at protein levels.

**Figure supplement 1—source data 1.** File containing original automatic capillary western blots (Simple Western assays) for *Figure 2—figure supplement 1A*, indicating the relevant bands and treatments.

**Figure supplement 1—source data 2.** Original files of capillary images of automatic capillary western blots (Simple Western assays) for *Figure 2—figure supplement 1A*.

**Figure supplement 1—source data 3.** File containing original automatic capillary western blots (Simple Western assays) for *Figure 2—figure supplement 1C*, indicating the relevant bands and treatments.

**Figure supplement 1—source data 4.** Original files of capillary images of automatic capillary western blots (Simple Western assays) for *Figure 2—figure supplement 1C*.

**Figure supplement 2.** Activity of different type of PKC inhibitors on spontaneous differentiation of suspension-cultured human induced pluripotent stem cells (hiPSCs).

spontaneous neuroectodermal differentiation and maintains the self-renewal of hiPSCs cultured in suspension conditions.

Next, we tested various PKC inhibitors to suppress neuroectodermal differentiation from hiPSCs in suspension conditions using another hiPSC line, 201B7 (*Figure 2—figure supplement 2A*). The suppression of *PAX6* expression in hiPSCs cultured in suspension conditions was observed with PKC inhibitors, enzastaurin (Ly317615), sotrastaurin (AEB071), Ro-32-0432, Gö6983, GF109203X, and LY333531, all of which possessed PKCβ inhibition activity (*Figure 2—figure supplement 2B–D*); however, sotrastaurin and Ro-32-0432 also showed growth inhibition of hiPSCs.

Further, we examined the expression pattern changes in specific isoforms of PKCs in hiPSCs cultured in adherent/suspension conditions. RNA expression of PKCα (PRKCA) and PKCβ (PRKCB) was significantly upregulated under suspension conditions compared to adhesion conditions (*Figure 3—figure supplement 1A*). Moreover, phosphorylated PKCβ protein expression was significantly elevated (*Figure 3—figure supplement 1B and C*). These results suggest that elevated expression and activation of PKCβ in suspension-cultured hiPSCs could affect the spontaneous differentiation.

## Combination of inhibitors of PKCβ and Wnt signaling pathways efficiently maintains self-renewal of hiPSCs in suspension conditions

To further explore the possibility that the inhibition of PKCβ is critical for the maintenance of self-renewal of hiPSCs in the suspension culture, we evaluated the effect of LY333531, a specific PKCβ inhibitor (*Jirousek et al., 1996*). Compared to controls, hiPSCs (WTC11 line) cultured in suspension conditions treated with IWR-1-endo and LY333531 formed homogeneous, round, smooth-surfaced aggregates (*Figure 3A*). *PAX6* expression was strongly suppressed by the addition of LY333531 (*Figure 3B*). Furthermore, after adding IWR-1-endo, the inhibitory effect of *PAX6* expression was further enhanced, and simultaneously, *OCT4* expression was restored to the same level as in the adherent-culture. PAX6 protein expression was also suppressed in hiPSCs treated with IWR-1-endo and LY333531 in suspension conditions while its expression increased in suspension conditions with conventional culture medium compared to adherent conditions (*Figure 2—figure supplement 1C and D*). The ratio of TRA-1-60-positive cells was higher in suspension conditions supplemented with IWR-1-endo and LY333531 than in control conditions without these inhibitors (*Figure 2—figure supplement 1E and F*). These results indicate that the maintenance of suspension-cultured hiPSCs is specifically facilitated by the combination of PKCβ and Wnt signaling inhibition. To examine the reproducibility of the effect of the inhibition of PKCβ and Wnt signaling pathways on the maintenance of

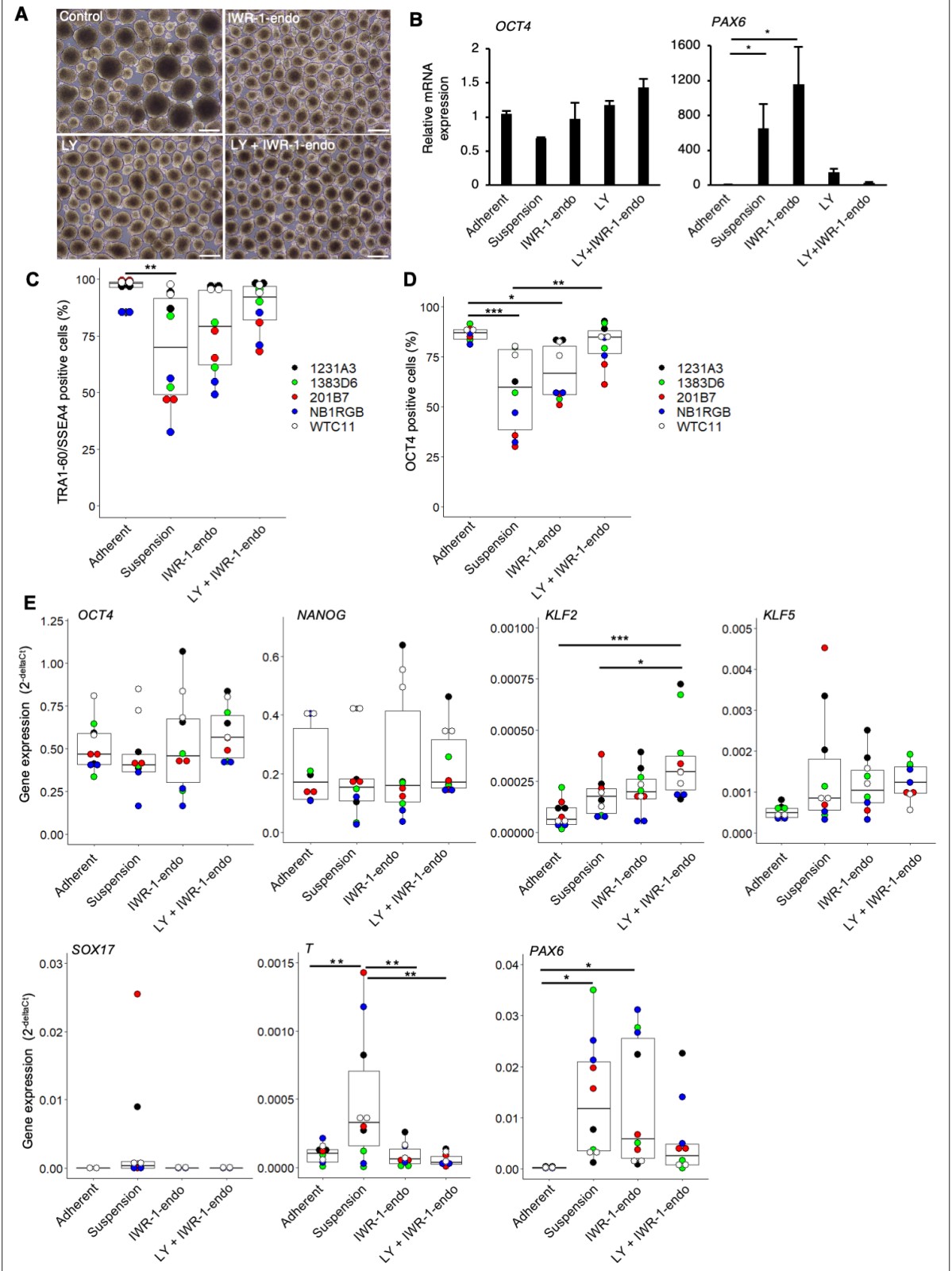

**Figure 3.** The inhibitors of Wnt and PKCβ efficiently maintain the self-renewal of human induced pluripotent stem cells (hiPSCs) in suspension conditions. (**A**) Phase-contrast images of suspension-cultured hiPSCs on day 10 (two passages) in the presence of IWR-1-endo, LY333531, or both. Scale bars: 400 μm. (**B**) Gene expression in suspension-cultured hiPSCs on P2 in the presence of IWR-1-endo, LY333531, or both. Data are presented as mean ± SE. p-values were statistically analyzed with Dunnett's test. (**C**) After suspension culture of five hiPSC lines, WTC11, 1231A3, HiPS-NB1RGB

*Figure 3 continued on next page*

*Figure 3 continued*

(NB1RGB), 1383D6, and 201B7, for 10 days (two passages) in StemFit AK02N medium with or without IWR-1-endo and LY333531, flow cytometry, immunocytochemistry, and RT-qPCR were performed. Box plots of flow cytometry for TRA-1-60 and SSEA4 double-positive cells (%) detected with flow cytometry are shown. Used cell lines are indicated by the different colored circles shown on the right side of the graph (n = 2 × 5 cell lines). (**D**) Box plots of OCT4-positive cells (%) detected with immunocytochemistry are shown (n = 2 × 5 cell lines). (**E**) Box plots of RT-qPCR data are shown. Undifferentiated markers (*OCT4, NANOG*), naïve pluripotency markers (*KLF2, KLF5*), and differentiation markers (*SOX17, T, PAX6*), were assessed. Statistical analysis was performed using one-way ANOVA and Tukey's test for all graphs. p-values <0.05 were considered statistically significant. *, **, or *** in the graphs indicate p<0.05, p<0.01, or p<0.001, respectively (n = 2 × 5 cell lines).

The online version of this article includes the following source data and figure supplement(s) for figure 3:

**Figure supplement 1.** Upregulation of PKC genes in suspension-cultured human induced pluripotent stem cells (hiPSCs).

**Figure supplement 1—source data 1.** File containing original automatic capillary western blots (Simple Western assays) for *Figure 3—figure supplement 1B*, indicating the relevant bands and culture conditions.

**Figure supplement 1—source data 2.** Original files of capillary images of automatic capillary western blots (Simple Western assays) for *Figure 3—figure supplement 1B*.

**Figure supplement 2.** Cell death of suspension-cultured human induced pluripotent stem cells (hiPSCs) after the treatment with LY333531 and IWR-1-endo.

the self-renewal of hiPSCs in suspension conditions among various hiPSCs, we evaluated the expression of self-renewal and differentiation markers among five different hiPSC lines, 1231A3, 1383D6, 201B7, HiPS-NB1RGB, and WTC11, simultaneously. Compared to adherent conditions, hiPSCs cultured in suspension conditions without chemical treatment decreased the positive ratio of TRA-1-60/SSEA4 and OCT4 and increased the expression levels of differentiation markers, *SOX17*, *T*, and *PAX6* (*Figure 3C–E*). These results indicate that suspension conditions without chemical treatment are unstable to maintain self-renewal and contain spontaneously differentiated cells. The addition of PKCβ and Wnt signal inhibitors increased the positive ratio of TRA-1-60/SSEA4 and OCT4 and decreased the expression levels of *SOX17*, *T*, and *PAX6* to the comparable level of adherent conditions. Interestingly, the expression of KLF2 and KLF5, which were known as naïve pluripotency markers, was upregulated in the suspension conditions treated with PKCβ and Wnt signal inhibitors. We also examined whether the combination of PKCβ and Wnt signaling inhibition affects cell survival in suspension conditions. In this experiment, we used another PKC inhibitor, Staurosporine (*Omura et al., 1977*), which has a strong cytotoxic effect as a positive control of cell death in suspension conditions. The addition of IWR-1-endo and LY333531 for 10 days had no effects on the apoptosis while the addition of Staurosporine for 2 hours induced Annexin-V-positive apoptotic cells (*Figure 3—figure supplement 2A–D*). These results indicate that the combination of PKCβ and Wnt signaling inhibition has no or little effects on the cell survival in suspension conditions. We next performed long-term culture for 10 passages in suspension conditions and compared hiPSC growth in the presence of LY333531 or Gö6983. When hiPSCs were seeded at $4 × 10^5$ cells/well, the average cell number reached approximately 12-fold after 5 days under both conditions (*Figure 4A and B*). After 10 passages, aggregates of hiPSCs cultured in the presence of LY333531 showed a uniform spherical shape, whereas aggregates cultured in the presence of Gö6983 were heterogeneously spherical (*Figure 4C*). Notably, in LY333531-treated cells, OCT4-positive cell numbers were significantly higher than in Gö6983-treated samples, as determined by immunostaining (*Figure 4D and E*). To evaluate whether hiPSCs cultured in suspension conditions with PKCβ and Wnt signaling inhibitors for 10 passages maintain pluripotency, we performed embryoid body (EB) formation assay. These EBs contained positive cells for TUJ1, SMA, and AFP as ectodermal, mesodermal, and endodermal marker, respectively (*Figure 4F*). Copy number variation (CNV) array analysis showed that hiPSCs cultured long term in the presence of PKCβ and Wnt inhibitors retained their normal human karyotype (*Figure 4G*). These results indicate that, for long-term culture, the inhibition of Wnt signaling and PKCβ in suspension conditions is sufficient to maintain the self-renewal, pluripotency, and genomic integrity of hiPSCs. Thus, we used the combination of IWR-1-endo and LY333531 for the rest of this study. We further investigated whether the effects of PKCβ and Wnt inhibitors on suppressing hiPSCs spontaneous differentiation in suspension conditions are applicable to other culture media. First, morphologies and gene expression profiles of hiPSCs cultured in suspension conditions with another commercially available maintenance medium, StemScale (Thermo Fisher Scientific, MA), were examined (*Figure 4—figure supplement 1A*). An hiPSC line, WTC11, cultured in suspension conditions treated with IWR-1-endo and LY333531 formed

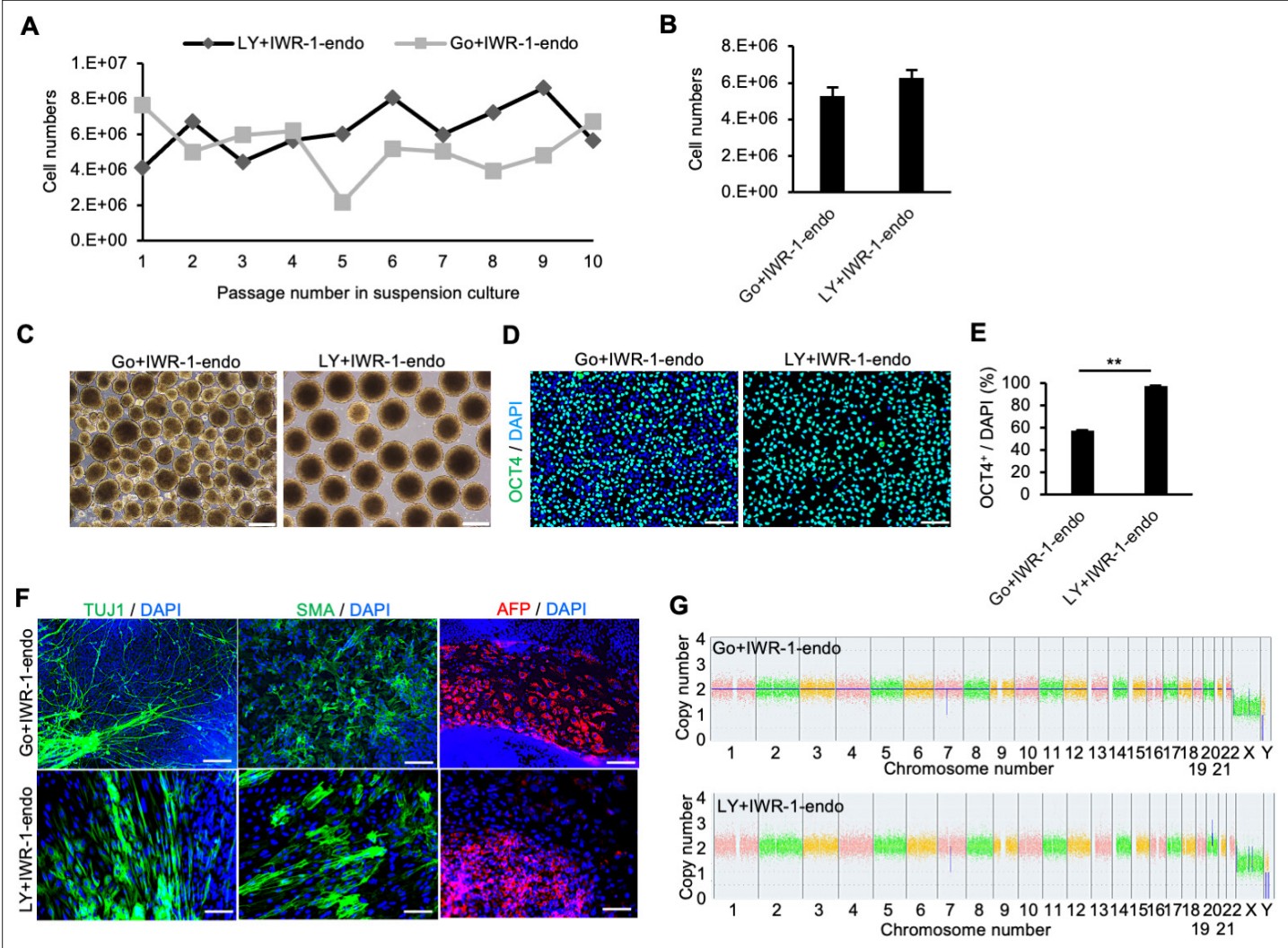

**Figure 4.** Long-term suspension culture of human induced pluripotent stem cells (hiPSCs) is maintained by simultaneous suppression of PKCβ and Wnt signals. (**A**) The number of hiPSCs (WTC11 line) counted at each passage (every 5 days) during long-term suspension culture. Cell culture was performed in the presence of IWR-1-endo plus Gö6983 or LY333531. (**B**) Bar graph indicating average cell numbers for 10 passages (mean ± SE). (**C**) Phase-contrast images of suspension-cultured hiPSCs on passage 10. Scale bars: 400 μm. (**D**) Immunocytochemistry of OCT4 on passage 10 samples. Scale bars: 100 μm. (**E**) Bar graph showing the percentages of OCT4-positive cells. Values were calculated from randomly selected three regions from immunofluorescence images. Data are presented as mean ± SE. p-value was statistically analyzed with Student's *t*-test. (**F**) Immunocytochemistry of differentiated cells in embryoid bodies (EBs) from suspension-cultured hiPSCs in the presence of IWR-1-endo, and Gö6983 or LY333531. Anti-TUJ1, -SMA, and -AFP antibodies were used to detect ectoderm, mesoderm, and endoderm differentiation, respectively. Scale bars: 100 μm. (**G**) Chromosomal copy numbers detected with copy number variation (CNV) array analysis (Karyostat assay) of suspension-cultured hiPSCs in the presence of IWR-1-endo and Gö6983 (upper panel) or LY333531 (lower panel) at passage 10. ** in the graphs indicate p<0.01.

The online version of this article includes the following figure supplement(s) for figure 4:

**Figure supplement 1.** Generality of the inhibitory effects of IWR-1-endo and LY333531 on spontaneous differentiation of human induced pluripotent stem cells (hiPSCs) cultured in suspension condition.

**Figure supplement 2.** Suspension culture of multiple human induced pluripotent stem cell (hiPSC) lines in mTeSR1 medium supplemented with or without IWR-1-endo and LY333531.

homogeneously round and smooth-surfaced aggregates compared to controls (*Figure 4—figure supplement 1B*). The differentiation markers expression, which was elevated under suspension conditions, was suppressed by the simultaneous addition of PKCβ and Wnt signal inhibitors, as observed with StemFit medium (*Figure 4—figure supplement 1C and D*). Second, morphologies and gene expression profiles of hiPSCs, cultured in suspension conditions with another commercially available maintenance medium, mTeSR1 (STEMCELL Technologies, Vancouver, Canada), were examined in four

different hiPSC lines (1231A3, 201B7, HiPS-NB1RGB, and WTC11) simultaneously (*Figure 4—figure supplement 2A*). Compared to adherent culture conditions, hiPSCs cultured in suspension conditions without chemical treatment significantly decreased the positive ratio of TRA-1-60/SSEA4 and OCT4 and increased the expression levels of differentiation markers, *SOX17*, *T*, and *PAX6* (*Figure 4—figure supplement 2B–D*). These results indicated that suspension conditions using mTeSR1 medium contain spontaneous differentiated cells. The addition of PKCβ and Wnt signal inhibitors increased the positive ratio of TRA-1-60/SSEA4 and OCT4 and decreased the expression levels of SOX17, T, and PAX6 to the comparable level of adherent conditions. Together, these results suggest that conventional suspension conditions contain spontaneous differentiating cells and that the addition of inhibitors of PKCβ and Wnt signaling pathways to conventional culture media generally suppresses spontaneous differentiation and maintains self-renewal.

## Global gene expression signatures in hPSCs supplemented inhibitors of PKCβ and Wnt signaling pathways in suspension conditions

We next examined the effect of PKCβ and Wnt inhibitors on global gene expression of hiPSCs under suspension conditions. Bulk RNA-seq data were obtained from suspension conditions in the absence of inhibitors (Sus), supplemented with IWR-1-endo (IWR), LY333531 (LY), IWR-1-endo and LY333531 (IWRLY), and adherent conditions (Ad). Hierarchical clustering obtained from these data showed that LY and IWRLY were grouped closely with Ad (*Figure 5A*). In contrast, Sus and IWR were both grouped as discrete populations from Ad. Additionally, hierarchical clustering in gene expression among these conditions was supported by principal component 1 (PC1) in the principal component analysis (PCA) (*Figure 5B*). In contrast, PC2 represented genes related to the effects of specific inhibitors under these conditions. Next, we investigated the effect of LY333531 and IWR-1-endo in suspension conditions. Many genes involved in pluripotency, *KLF4* and *ID1*, and epithelial cell–cell interactions, *CDH1* (E-cadherin), were significantly upregulated in IWRLY, while many transcription factors involved in differentiation—*PAX2*, *PAX3*, *PAX5*, *PAX8*, *SP5*, *DBX1*, and *TFAP2B*—were downregulated in IWRLY (*Figure 5C*). GSEA and GOEA on downregulated genes in IWRLY showed that the expression of developmentally associated genes, whose expression was elevated in Sus, was generally reduced in IWRLY (*Figure 5D and E*). GOEA on upregulated genes revealed gene sets involved in epithelial cell types (*Figure 5F*). Compared to Ad, genes involved in sensory system development, cell–cell adhesion, and Wnt and PI3K signaling pathways were upregulated in IWRLY, and genes involved in nucleotide metabolism and hypoxic responses were downregulated under IWRLY conditions (*Figure 5—figure supplement 1A–D*). These results suggest that PKCβ and Wnt signaling inhibitors in suspension conditions regulate global gene expression patterns to suppress spontaneous differentiation, albeit remaining expression signatures of suspension culture, possibly due to the microenvironment within the formed aggregates and physiological differences. We also extracted and analyzed individual gene expression data of pluripotency markers from RNA-seq results. Compared to adherent conditions, the expression of naïve pluripotency markers, *KLF2*, *KLF4*, *KLF5*, and *DPPA3*, was upregulated in IWRLY conditions while *OCT4* and *NANOG* were at the similar levels (*Figure 5—figure supplement 2*). Combined with RT-qPCR analysis data on five different hiPSC lines (*Figure 3E*), these results suggest that IWRLY conditions may drive hiPSCs to shift toward naïve pluripotent states in suspension conditions.

## Mass expansion of hiPSCs in suspension conditions supplemented with inhibitors of PKCβ and Wnt signaling pathways

For the clinical applications of hiPSCs, its homogeneous mass production is required to obtain sufficient quantities. To test the feasibility of mass production under suspension conditions supplemented with PKCβ and Wnt signal inhibitors, we first performed suspension culture using a healthy donor-derived hiPSC line, 1383D6, in a 30 mL bioreactor with stirring conditions (*Matsumoto et al., 2022*) at different cell seeding densities and different stirring speeds (*Figure 6—figure supplement 1A*). At 150 rpm of stirring speed, hiPSCs steadily proliferated at $0.5 \times 10^5$–$2 \times 10^5$ cells/mL of the cell seeding density, but cells hardly proliferated at $8 \times 10^5$ cells/mL (*Figure 6—figure supplement 1B*). Since the number of total collected cells was the lowest at a seeding density of $0.5 \times 10^5$ cells/mL, the seeding density of $1 \times 10^5$–$2 \times 10^5$ cells/mL is considered suitable. Also, at the seeding density of $2 \times 10^5$ cells/mL, hiPSCs steadily proliferated at 50–150 rpm of the stirring speed, but not at 250 rpm (*Figure 6—figure supplement 1C*). Then, we analyzed protein expression of PAX6 and SOX17 in these cells after

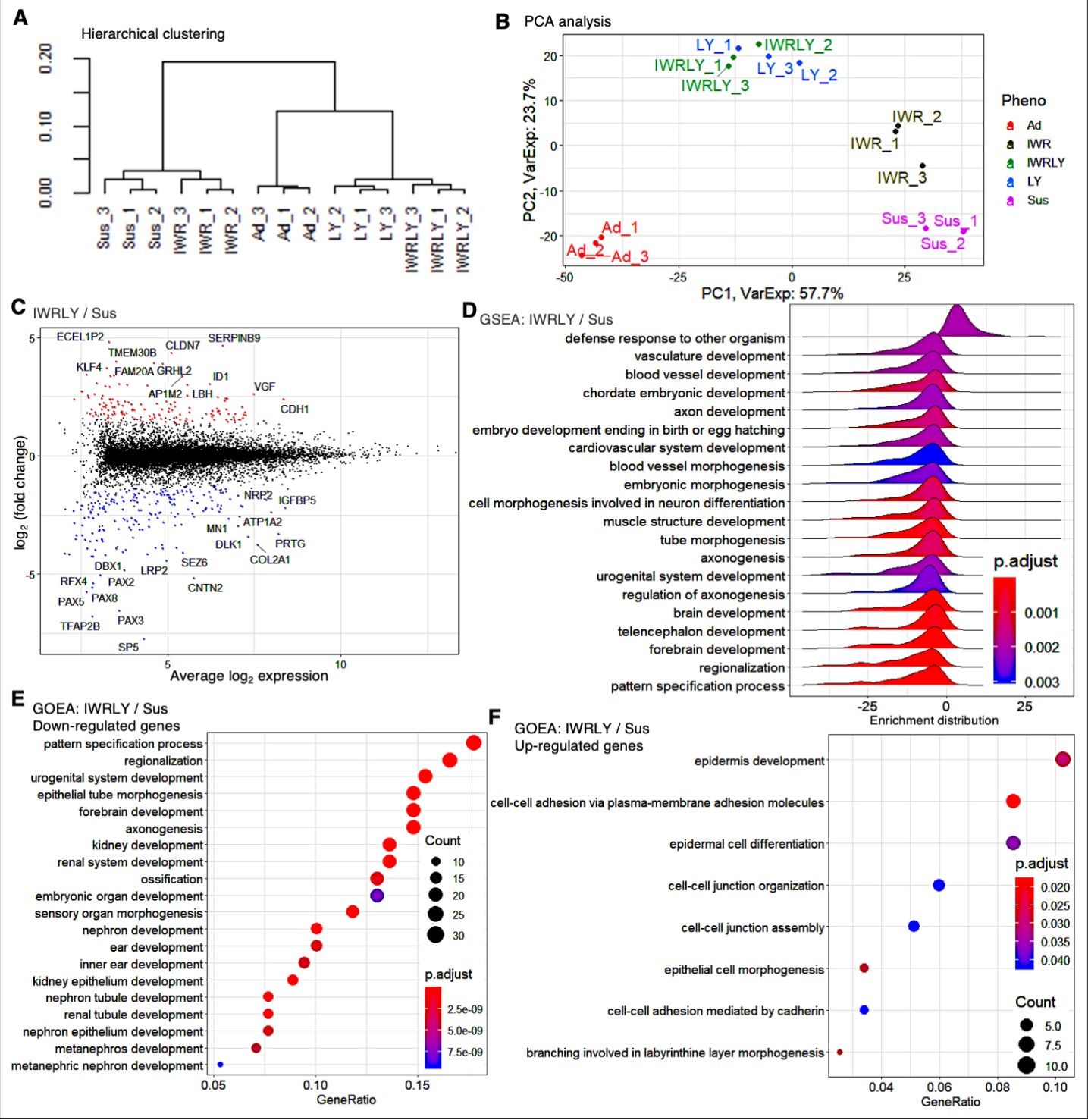

**Figure 5.** The inhibitors of Wnt and PKCβ in suspension conditions efficiently suppress differentiated gene marker expression in transcriptome analysis. (**A**) Hierarchical clustering of adherent and suspension-cultured human induced pluripotent stem cells (hiPSCs) (WTC11 line) on P2 using Ward's method from RNA-seq data (n = 3 in each condition). Ad, adherent; Sus, suspension; IWR, IWR-1-endo; LY, LY333531; IWRLY, IWR-1-endo, and LY333531. (**B**) Principal component analysis (PCA) plot showing clusters of samples based on similarity. Gene expression variance are displayed as PC1 and PC2. (**C**) MA plot (log2 fold change versus mean average expression) comparing transcriptomes between IWRLY and Sus conditions. (**D**) Gene Set Enrichment Analysis (GSEA) on the gene sets of IWRLY to Sus from these RNA-seq data. Statistically significant enrichment is shown. p-values are represented in blue to red from low to high values. (**E, F**) Gene Ontology Enrichment Analysis (GOEA) for the gene sets of IWRLY to Sus from these RNA-seq data. Analysis was performed on downregulated genes in (**E**) and upregulated genes in (**F**).

*Figure 5 continued on next page*

*Figure 5 continued*

The online version of this article includes the following figure supplement(s) for figure 5:

**Figure supplement 1.** Global gene expression of suspension-cultured human induced pluripotent stem cells (hiPSCs) with IWR-1-endo and LY333531 in comparison to adherent-cultured hiPSCs.

**Figure supplement 2.** Comparison of expression on naïve pluripotency markers between adherent and suspension-cultured human induced pluripotent stem cells (hiPSCs).

three passages with these conditions at 50–150 rpm of stirring speed. The addition of PKCβ and Wnt inhibitors decreased the positive ratio of PAX6 and SOX17 in these reactor stirring speeds (*Figure 6—figure supplement 1D*). These results suggest that PKCβ and Wnt inhibitors suppressed spontaneous differentiation in bioreactor conditions at suitable stirring speeds.

Next, we examined that undifferentiated hiPSCs were efficiently maintained in these bioreactor conditions with five serial passages using two hiPSC lines, 1383D6 and 1231A3 (*Figure 6—figure supplement 2A*). These cell lines showed higher cell density in the bioreactor at each passage after 3 days (*Figure 6—figure supplement 2B*). Under these conditions, the concentration of glucose decreased daily (*Figure 6—figure supplement 2C*), whereas that of L-lactic acid increased (*Figure 6—figure supplement 2D*). These results indicate that the cell lines proliferated with active energy consumption. Further, RT-qPCR analysis showed no marked differences in the undifferentiated and differentiation markers expression between hiPSCs cultured in adhesion conditions and suspension conditions with PKCβ and Wnt inhibitors (*Figure 6—figure supplement 2E and F*). Flow cytometric analysis showed that more than 90% of these cells were positive for pluripotency marker proteins: NANOG, OCT4, and SOX2 (*Figure 6—figure supplement 2G*). G-band analysis of suspension-cultured hiPSCs after five passages revealed normal karyotype (*Figure 6—figure supplement 2H*). These results demonstrate that undifferentiated hiPSCs were efficiently maintained in the bioreactor using the culture medium supplemented with inhibitors of PKCβ and Wnt signaling pathways.

Then, we expanded hiPSCs in a large-scale culture system under perfusion conditions in the presence of IWR-1-endo and LY333531. In this experiment, a clinical-grade hiPSC line, Ff-I14s04, which is derived from peripheral blood mononuclear cells (PBMCs) of a donor carrying homozygous alleles for major HLA loci (HLA-A, HLA-B, and HLA-DR), was used (*Kitano et al., 2022*). Large-scale hiPSCs preparation using a perfusion-culture system with 320 mL bioreactor having stirred wing (*Kropp et al., 2016*) was performed in GMP-compliant, clinical-grade StemFit AK03N medium containing IWR-1-endo and LY333531 (*Figure 6A*). When the culture was started at $1 \times 10^5$ cells/mL in 320 mL medium scale, hiPSCs proliferated approximately tenfold after 3–4 days to produce ~300 stock vials ($1 \times 10^6$ cells/vial). This large-scale culture was repeated three times (passages 1–3). Since the population doubling time (PDT) of this hiPSC line in adherent culture conditions is 21.8–32.9 hours measured at its production, the proliferation rate in this large-scale suspension culture is comparable to adherent culture conditions. Next, the frozen vials of this hiPSC line obtained at each passage in large-scale suspension culture conditions were characterized. After these vials were thawed and seeded in adherent-culture conditions, these cells formed typical hiPSC-like colonies (*Figure 6B*). These samples from suspension conditions showed similar or higher viability (>90%) to that of adherent-culture-derived vials (*Figure 6C*). When compared to adherent conditions, these samples from suspension conditions showed a similar or higher proliferation rate after thawing (*Figure 6D*). Flow cytometric analysis showed that over 90% of cells were positive for pluripotent cell markers expression: TRA-1-60, SSEA4, and OCT4 (*Figure 6E and F*). Further, G-band analysis revealed that hiPSCs retained their normal karyotype even after three passages under large-scale suspension conditions (*Figure 6G*). When large-scale suspension-cultured hiPSCs were incubated in each germ layer-specific differentiation medium for 4–7 days, the expression of early differentiation markers for ectoderm (*PAX6* and *SOX1*), mesoderm (*T* and *PDGFRA*), and endoderm (*SOX17* and *CXCR4*) was significantly induced (*Figure 6H*). These cells were then directly differentiated into dopaminergic neural progenitors, cardiomyocytes, and hepatocytes to evaluate their differentiation capacity and propensity. There were no differences in the differentiation efficiency toward these lineages (*Figure 6I–M*). These results indicate that the characteristics and quality of clinical-grade hiPSCs cultured in large-scale suspension conditions in the presence of PKCβ and Wnt inhibitors are equivalent to those of hiPSCs maintained under adherent conditions. Taken together, we were successful in mass suspension conditions of hiPSCs supplemented with Wnt and PKCβ inhibitors.

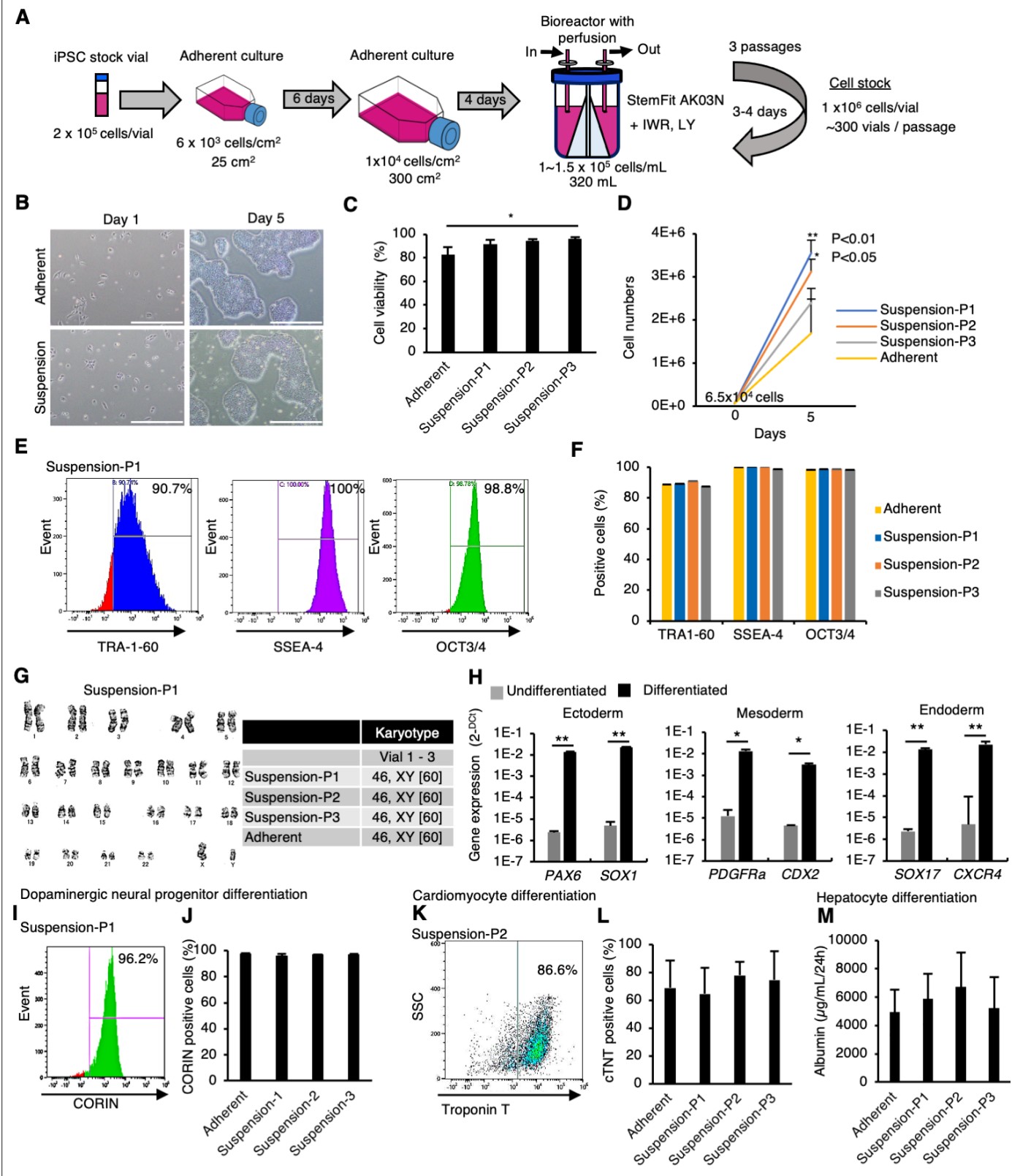

**Figure 6.** Mass suspension culture of clinical-grade human induced pluripotent stem cells (hiPSCs) in the presence of PKCβ and Wnt signaling inhibitors. (**A**) Schematics of mass suspension culture using a bioreactor. (**B**) Representative phase-contrast images of hiPSCs after seeding from frozen vials. Scale bar: 1 mm. (**C**) Cell viability at seeding. Data are presented as mean ± SE (n = 3). p-values were statistically analyzed with Dunnett's multiple comparisons test. (**D**) Total cell numbers were counted on day 5 after thawing. Data were presented as mean ± SE (n = 3). p-values were statistically

*Figure 6 continued on next page*

*Figure 6 continued*

analyzed with Dunnett's test. (**E**) Representative flow cytometry data of pluripotent markers in these hiPSCs. (**F**) Quantification of flow cytometry data for the pluripotent markers. Data are presented as mean ±SE (n = 3). (**G**) Karyotypes of these hiPSCs. Left: representative pictures of G-band analysis. Right: table of karyotype results (n = 3). The numbers in brackets indicate the cell numbers examined. (**H**) In vitro differentiation into early phases of three germ layers from hiPSCs was assessed using RT-qPCR (mean ± SE) (n = 3). p-value was statistically analyzed with Student's *t*-test. (**I**) Representative flow cytometry data for CORIN in the cells in the dopaminergic neural progenitors differentiated from hiPSCs. (**J**) Quantification of CORIN-positive cells (mean ± SE, n = 3). (**K**) Representative flow cytometry data for cardiac Troponin T (cTnT) in the cardiomyocytes differentiated from hiPSCs. (**L**) Quantification of cTnT-positive cells (mean ± SE, n = 3). (**M**) Albumin secretion levels of hepatocytes differentiated from hiPSCs (mean ± SE, n = 3). * or ** in the graphs indicate p<0.05 or p<0.01, respectively.

The online version of this article includes the following figure supplement(s) for figure 6:

**Figure supplement 1.** Suspension culture of human induced pluripotent stem cells (hiPSCs) supplemented with IWR-1-endo and LY333531 using bioreactors with continuous agitation.

**Figure supplement 2.** Suspension culture of human induced pluripotent stem cells (hiPSCs) supplemented with IWR-1-endo and LY333531 using bioreactors with continuous agitation.

## Single-cell sorting and expansion of hiPSC subclones cultured in suspension conditions

To test the feasibility of our suspension culture method to single-cell sorting, we sorted a hiPSC line, 201B7, with TRA-1-60 antibody into individual wells of a 96-well plate and expanded them with serial passages using StemFit AK02N medium (*Figure 7A*). On day 7 after single-cell sorting, we counted the number of colonies to calculate the cloning efficiency (*Figure 7B*). The cloning efficiency in adherent culture conditions was approximately 30%. While the cloning efficiency in suspension conditions without any chemical treatment was less than 10%. The treatment of IWR-1-endo in the suspension-culture conditions increased the efficiency more than 20%, although the treatment of LY333531 decreased the efficiency. These results indicate that the IWR-1-endo treatment is beneficial in single-cell cloning in suspension conditions. On day 14, we performed passages of single-cell-derived colonies and cultured in suspension conditions supplemented with IWR-1-endo LY333531. By day 28 after this single-cell sorting, we expanded single-cell-derived hiPSC subclones in suspension culture supplemented with IWR-1-endo and LY3333531. On day 28, we examined seven subclones for their cell growth and expression of OCT4 and TRA-1-60. The subclones showed round-shaped aggregates with more than 3 million cells and high ratios of OCT4- and TRA1-60-positive cells (*Figure 7C–F*). These results indicate that we have successfully derived single-cell-cloned sublines in suspension conditions.

## Direct freeze and thaw of hiPSCs cultured in suspension conditions

To test the feasibility of our suspension culture method to direct freeze and thaw processes, we froze a hiPSC line, 201B7, in suspension conditions using StemFit AK02N medium supplemented with IWR-1-endo and LY3333531. Then, we thawed these frozen vials and directly reseeded the cells in suspension conditions supplemented with IWR-1-endo and LY3333531 (*Figure 8A*). By day 10 after reseeding, we expanded the hiPSCs in suspension conditions supplemented with IWR-1-endo and LY3333531 (*Figure 8B*). On day 10, we examined three vials for their cell growth and expression of OCT4 and TRA-1-60. The subclones showed more than 3 million cells and high ratios of OCT4- and TRA1-60-positive cells (*Figure 8C–E*). We also tested mTeSR1 medium for this process. Three different hiPSC lines, WTC11, 1231A3, and HiPS-NB1RGB, cultured in mTeSR1 supplemented with IWR-1-endo and LY3333531 successfully recovered from frozen vials in suspension conditions supplemented with IWR-1-endo and LY3333531 (*Figure 8—figure supplement 1A–D*). These results indicate that we have successfully frozen and thawed hiPSCs in suspension conditions directly.

## Establishment of hiPSC lines in complete suspension conditions supplemented with inhibitors of PKCβ and Wnt signaling pathways

Finally, we aimed to establish hiPSCs in suspension conditions. Using human PBMCs as a starting material, we generated hiPSCs using a novel replication-defective and persistent Sendai virus vector (SeVdp) infection or episomal vector electroporation as these methods are well known for producing transgene-free, clinical-grade hiPSCs (*Fusaki et al., 2009*; *Nishimura et al., 2011*; *Figure 9A*). PBMCs

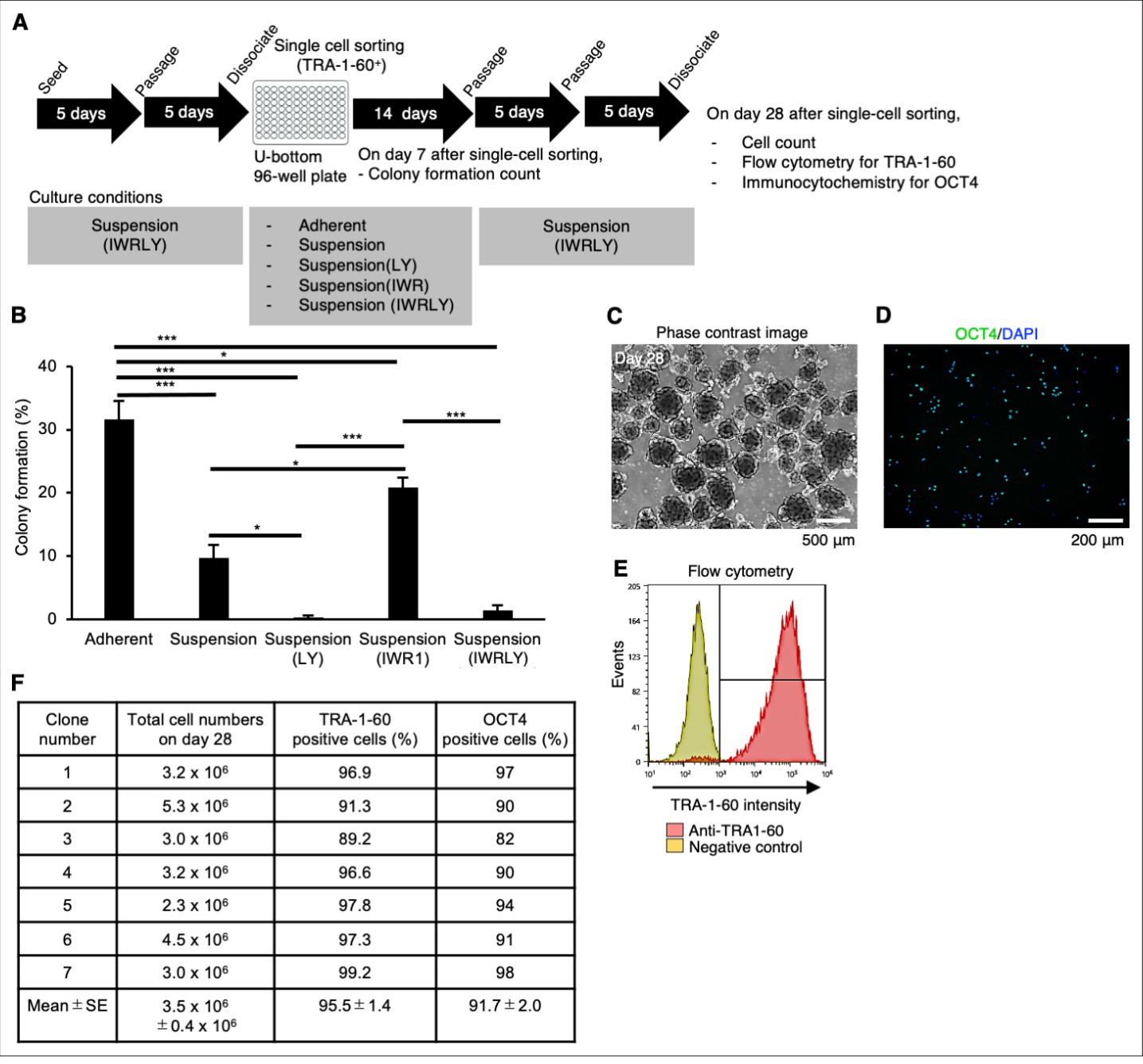

**Figure 7.** Establishment of single-cell-sorted human induced pluripotent stem cell (hiPSC) subclones cultured in suspension conditions supplemented with IWR-1-endo and LY3333531. (**A**) Schematics representing the establishment of single-cell-derived hiPSC subclones from 201B7 line. Single-cell-sorted cells were expanded in the culture medium supplemented with IWR-1-endo and LY333531 (IWRLY). Formed colonies were picked on day 14, and expanded by repeating passage every 4–5 days under suspension conditions. Characteristic analysis was performed on day 28 after single-cell sorting. (**B**) On day 7 after single-cell sorting, formed colonies were counted per well in the 96-well plate. The ratio (%) of the colony formation is shown in a bar graph (mean ± SE) (n = 3). Statistical analysis was performed using one-way ANOVA and Tukey's tests for all graphs. p-values <0.05 were considered statistically significant. *, **, or *** in the graphs indicate p<0.05, p<0.01, or p<0.001, respectively. (**C**) Phase-contrast images of Clone #2 on day 28 after single-cell sorting. Scale bars: 500 µm. (**D**) Represented immunocytochemistry of OCT4 (Clone #2). Scale bars: 200 µm. (**E**) Represented flow cytometry of TRA-1-60 (Clone #2). (**F**) Summary table of the characterization of single cell-sorted clones. Total cell numbers on day 28, the ratio of TRA-1-60-positive cells (%), and the ratio of OCT4-positive cells (%) are shown (mean ± SE) (n = 7).

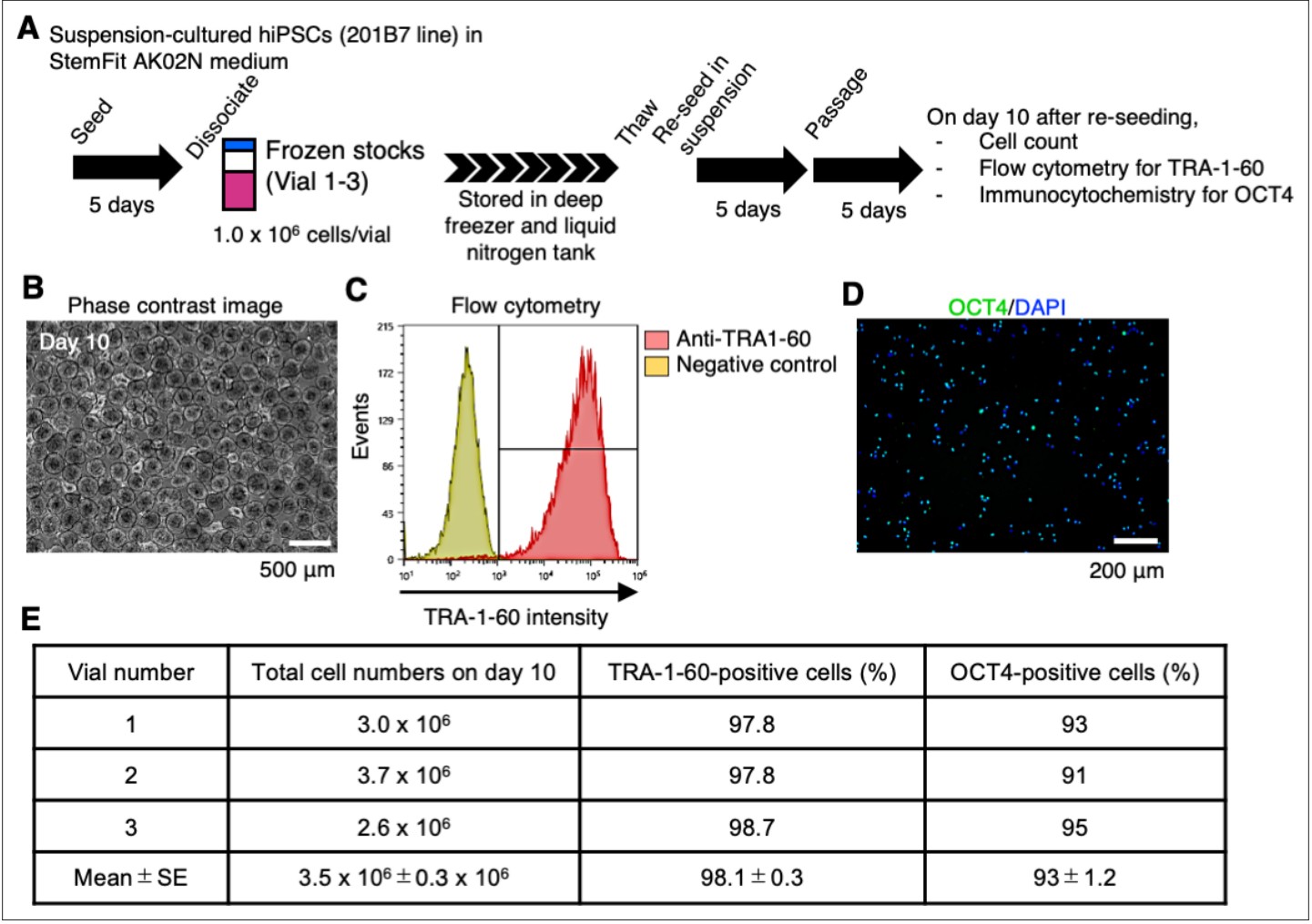

**Figure 8.** Direct re-suspension culture from frozen stocks of human induced pluripotent stem cells (hiPSCs) in suspension conditions supplemented with IWR-1-endo and LY3333531. (**A**) Schematics of direct re-suspension culture of frozen stocks of single-cell-derived 201B7 clone (generated in *Figure 7*; Vials 1-3). (**B**) Represented phase-contrast images on day 10 (Vial 3). Scale bars: 500 µm. (**C**) Represented flow cytometry of TRA-1-60 (Vial 3). (**D**) Represented immunocytochemistry of OCT4 (Vial 3). Scale bars: 200 µm. (**E**) Summary table of the characterization of re-suspension-cultured hiPSCs from frozen stocks. Total cell numbers on day 10, the ratio of TRA-1-60-positive cells (%), and the ratio of OCT4-positive cells (%) are shown (mean ± SE) (n = 3).

The online version of this article includes the following figure supplement(s) for figure 8:

**Figure supplement 1.** Direct re-suspension from frozen stocks of human induced pluripotent stem cells (hiPSCs) under suspension conditions supplemented with IWR-1-endo and LY333531 in mTeSR1 medium.

were infected with SeVdp carrying *OCT4*, *SOX2*, *KLF4*, and *L-MYC* genes. Infected cells were cultured by repeating passages every 5–6 days. Cell aggregates cultured with IWR-1-endo and LY333531 showed uniform spherical structure (*Figure 9B*), and most of the cells were positive for OCT4 (*Figure 9C*) and TRA-1-60 (*Figure 9D*) on day 56. These bulk reprogrammed cells were able to differentiate into three germ layers in an in vitro EB formation assay (*Figure 9E*) and in teratomas that were transplanted into immunodeficient NOD.Cg-Prkdcscid Il2rgtm1Wjl/SzJ (NSG) mice (*Figure 9F*). These cells maintained normal karyotype (*Figure 9G*). These results demonstrate that these bulk cells were efficiently reprogrammed to hiPSCs without any sorting or selecting procedures. We then applied these bulk hiPSCs for single-cell sorting with fluorescent-labeled TRA-1-60 antibody and expanded single-cell-derived clones in suspension conditions. A clone (F-10) showed OCT4 and TRA1-60 expressions (*Figure 9H–J*). In addition, the established clone showed potency to differentiate into derivatives of three germ layers in vitro as EBs (*Figure 9K*) and in vivo as teratoma transplanted into NSG mice (*Figure 9L*). A normal karyotype was observed in this clone (*Figure 9M*). SeVdp was nearly

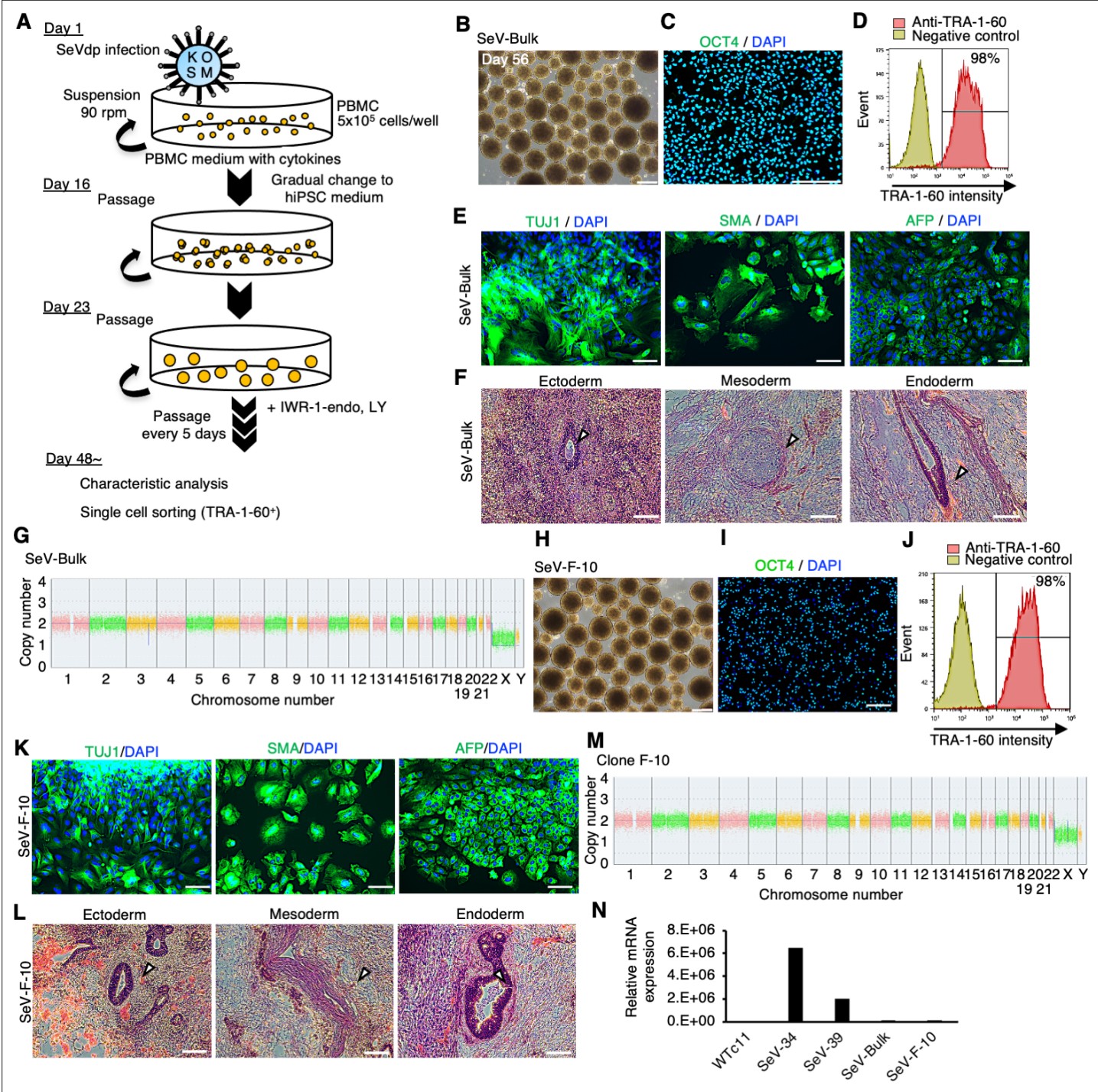

**Figure 9.** Establishment of human induced pluripotent stem cells (hiPSCs) in complete suspension conditions using SeVdp. (**A**) Schematics of hiPSC generation in suspension conditions. (**B**) Phase-contrast images of peripheral blood mononuclear cells (PBMCs) on day 56 after infection. Scale bars: 400 μm. (**C**) Immunocytochemistry of OCT4 on bulk-hiPSCs on day 56. Scale bars: 200 μm. (**D**) Flow cytometry of TRA-1-60 in bulk-hiPSCs on day 61. (**E**) Immunocytochemistry of TUJ1, SMA, and AFP on differentiated cells in embryoid bodies (EBs) from bulk-hiPSCs on day 56. Scale bars: 100 μm. (**F**) HE staining of teratoma sections derived from bulk-hiPSCs. White arrowheads indicate representative tissue structures derived from ectoderm, mesoderm, and endoderm. Scale bars: 100 μm. (**G**) Chromosomal copy numbers detected with copy number variation (CNV) array analysis on bulk-hiPSCs. (**H**) Phase-contrast image of an established hiPSC clone at passage 7. Scale bars: 400 μm. (**I**) Immunocytochemistry of established clone, F-10, with anti-OCT4 antibody. Scale bars: 200 μm. (**J**) The percentage of TRA-1-60-positive cells with flow cytometry on established clone F-10 line. (**K**) HE staining of teratoma sections derived from F-10 clone line. The details are the same as in (**F**). (**L**) Immunocytochemistry in EBs from established F-10 clone. Scale bars: 100 μm. (**M**) Chromosomal copy numbers of F-10 clone line. (**N**) Residual SeVdp genomic RNA in established hiPSCs with RT-qPCR.

*Figure 9 continued on next page*

*Figure 9 continued*

The online version of this article includes the following figure supplement(s) for figure 9:

**Figure supplement 1.** Establishment of human induced pluripotent stem cells (hiPSCs) in completed suspension conditions using episomal vectors.

extinct in these bulk populations and this clone (*Figure 9N*). These results demonstrate that we are successful in establishing transgene-free hiPSC lines using SeVdp infection in suspension conditions in the presence of PKCβ and Wnt inhibitors.

To repeat iPSC generation in suspension conditions in the presence of PKCβ and Wnt signaling inhibitors with different settings, we aimed to establish single-cell-derived hiPSC lines using the transfection of episomal plasmid vectors. PBMCs were transfected with episomal plasmid vector mix carrying *OCT4*, *SOX2*, *KLF4*, *L-MYC*, *LIN28*, and *mp53DD* and cultured in suspension conditions in the presence or absence of IWR-1-endo and LY33351 with repeated passages to reach enough cell numbers to make cell stocks and characterization (*Figure 9—figure supplement 1A*). After 36 days of culture, 59% double-positive cells for TRA-1-60 and SSEA4 were observed in control, whereas 78% were seen in the presence of IWR-1-endo and LY333531 (*Figure 9—figure supplement 1B and C*). We performed cell sorting of TRA-1-60 and SSEA4 double-positive cells and further repeated five more passages in the absence or presence of IWR-1-endo and LY333531. Cell aggregates cultured in culture medium supplemented with IWR-1-endo and LY333531 showed uniform spherical structures (*Figure 9—figure supplement 1D*), and most of these cells were TRA-1-60 and SSEA4 double-positive (*Figure 9—figure supplement 1E*). In contrast, cell aggregates cultured without these inhibitors showed lumpy, heterogeneous shapes, and decreased undifferentiated population. Immunostaining of OCT4 on dissociated cell aggregates resulted in higher OCT4-positive cells in IWR-1-endo and LY333531-treated cells (*Figure 9—figure supplement 1F and G*). We further performed single-cell sorting of TRA-1-60-positive cells from bulk-cultured populations at 10 passages (*Figure 9—figure supplement 1H and I*). We selected three clones to be characterized further. These clones showed round spherical shapes in their aggregates and expression of OCT4, NANOG, and TRA-1-60 proteins (*Figure 9—figure supplement 1J–L*). Pluripotency into three germ layers was confirmed in vitro with EB formation assay (*Figure 9—figure supplement 1M*) and teratoma formation analysis (*Figure 9—figure supplement 1N*). These clones retained a normal karyotype even after a long-culture period (*Figure 9—figure supplement 1O*). These results demonstrate that single-cell-derived hiPSC lines were successfully established using the transfection of episomal plasmid vectors in suspension conditions in the presence of PKCβ and Wnt signaling inhibitors.

## Discussion

In this study, we have developed a series of methods to generate and maintain hiPSCs in suspension conditions. First, we have identified compounds that suppress the spontaneous differentiation of hiPSCs in suspension conditions. Based on these findings, we have newly achieved a complete series of culture processes including hiPSC establishment, long-term culture, mass culture, single-cell cloning, and direct freeze and thaw. Our methods are validated in several conventional culture media and many hiPSC lines. Thus, our findings show that suspension conditions with Wnt and PKCβ inhibitors (IWRLY suspension conditions) can precisely control cell conditions and are comparable to conventional adhesion cultures regarding cellular function and proliferation. Many previous 3D culture methods intended for mass expansion used hydrogel-based encapsulation or microcarrier-based methods to provide scaffolds and biophysical modulation (*Chan et al., 2020*). These methods are useful in that they enable mass culture while maintaining scaffold dependence. However, the need for special materials and equipment and the labor and cost involved are concerns toward industrial mass culture of hiPSC-related products. On the other hand, our IWRLY suspension conditions do not require special materials such as hydrogels, microcarriers, or dialysis bags, and have the advantage that common bioreactors can be used. Furthermore, we have observed some differences in conventional media used for suspension-culture conditions in maintaining self-renewal characteristics, preventing spontaneous differentiation into specific lineages, and performing stability among different experimental times. Overcoming these heterogeneity caused by conventional media, the IWRLY suspension conditions robustly maintain hiPSC self-renewal and pluripotency. Therefore, this IWRLY suspension

conditions for hiPSCs is advantageous in terms of mass culture, automation, safety assurance, and is expected to have novel industrial applications that could not be achieved by conventional methods.

HiPSCs have been generally considered to be scaffold-dependent and are cultured under adherent monolayer culture conditions (*Hayashi and Furue, 2016*; *Xu et al., 2001*). On the other hand, some studies reported that floating cultures without external ECM addition or scaffolds have been successfully performed on hiPSCs for their long-term and/or mass expansion (summarized in *Table 1*). However, dissociation into single cells and amplification from these single cells in suspension conditions have never been achieved. Thus, it remains arguable whether culturing human pluripotent stem cells without providing a scaffold is possible. In this study, we demonstrated that many existing hiPSCs quickly acclimatized to the IWRLY suspensionconditions and that they could be successfully cultured even under harsh conditions such as colony formation from single cells or direct freeze–thaw processes. These findings suggest that hiPSCs can be sufficiently cultured even without scaffolds or exogenous ECM proteins. However, as previous studies have shown, hiPSCs themselves secrete scaffold substances such as ECM proteins, which may affect the status of suspension conditions (*Kim et al., 2019*).

It is interesting to see whether and how the properties of hiPSCs cultured in IWRLY suspension conditions are altered from the adherent conditions. Our transcriptome results in comparison to adherent conditions show that gene expression associated with cell-to-cell attachment, including E-cadherin (CDH1), is more activated. This may be due to the status that these hiPSCs are more dependent on cell-to-cell adhesion where there is no exogenous cell-to-substrate attachment in the three-dimensional culture. Previous studies have shown that cell-to-cell adhesion by E-cadherin positively regulates the survival, proliferation, and self-renewal of human pluripotent stem cells (*Aban et al., 2021*; *Li et al., 2012*; *Ohgushi et al., 2010*). Furthermore, studies have shown that human pluripotent stem cells can be cultured using an artificial substrate consisting of recombinant E-cadherin protein alone without any ECM proteins (*Nagaoka et al., 2010*). Also, cell-to-cell adhesion through gap junctions regulates the survival and proliferation of human pluripotent stem cells (*Wong et al., 2006*; *Wong et al., 2004*). These findings raise the possibility that the cell-to-cell adhesion, such as E-cadherin and gap junctions, is compensatory activated and supports hiPSC self-renewal in situations where there are no exogenous ECM components and its downstream integrin and focal adhesion signals are not forcedly activated in suspension conditions. It will be interesting to elucidate these molecular mechanisms related to E-cadherin and gap junctions in the hiPSC survival and self-renewal in IWRLY suspension conditions in the future.

We have identified two compounds as factors that ameliorate the disadvantages of the suspension conditions of hiPSCs. Wnt signaling inhibitors suppressed spontaneous differentiation toward mesendoderm, PKCβ inhibitors suppressed spontaneous ectodermal differentiation, and the combination of these two inhibitors maintained the undifferentiated nature of hiPSCs with high efficiency in suspension conditions. As for Wnt activation in human pluripotent stem cells, previous studies reported some Wnt agonists were expressed in undifferentiated human pluripotent stem cells (*Dziedzicka et al., 2021*; *Jiang et al., 2013*; *Konze et al., 2014*). In suspension conditions, cell aggregation causes tight cell–cell interaction. The autocrine and paracrine effects of Wnt agonists in the cell aggregation may strongly affect neighbor cells to induce spontaneous differentiation into mesendodermal cells. Thus, the inhibition of Wnt signaling should be effective in suppressing the spontaneous differentiation into mesendodermal lineages in suspension conditions. Gö6983, a pan-PKC inhibitor, has been used to promote the self-renewal of mammalian PSCs (*Dutta et al., 2011*; *Kinehara et al., 2013*; *Rajendran et al., 2013*; *Takashima et al., 2014*); however, in these cases, the role of specific isoforms of PKC in PSC self-renewal and differentiation is not fully elucidated yet. Also, PKC signaling is involved in neural induction conserved in animals shown in sea urchin (*Range et al., 2013*), *Xenopus* (*Otte et al., 1988*; *Otte et al., 1989*), and mouse (*Stumpo et al., 1995*). Our findings suggested the involvement of PKCβ in human neuroectodermal differentiation using hiPSCs. It will be interesting to elucidate the molecular mechanisms of how PKC signaling is involved in neuroectoderm differentiation in the future.

Interestingly, these combinations of chemical inhibitors against Wnt and PKC signaling pathways are also being used in the induction and maintenance of human naïve pluripotent stem cells (*Bayerl et al., 2021*; *Bredenkamp et al., 2019*; *Guo et al., 2017*; *Khan et al., 2021*). Previous studies suggested that cell state transition toward naïve state in hPSCs had beneficial effects in the suspension conditions of hiPSCs (*Lipsitz et al., 2018*; *Rohani et al., 2020*). While we did not aim to drive

hiPSCs to a naïve state by the use of these chemical inhibitors in suspension conditions, we found that naïve pluripotency marker genes were upregulated in the IWRLY suspension conditions consistently. The question of why the transition to the naïve PSCs type facilitates the culture adaptation to suspension conditions remains elusive; however, considering early development, the inner cell mass in blastocysts in the earlier stages that maintains pluripotency forms a three-dimensional morphology, whereas epiblasts, where the development is more proceeded, form a flattened epithelial morphology (*Sheng, 2015*). The same is true for pluripotent stem cells in mice, where naïve mouse embryonic stem cells derived from blastocysts form three-dimensional colonies, whereas epiblast stem cells derived from the later developmental stages form flattened epithelial colonies (*Brons et al., 2007*; *Tesar et al., 2007*). We previously showed that these differences in the morphology of mouse pluripotent stem cells could be regulated by the activation state of ECM and integrin signaling and that the situation where these signals are not active is suited to the naïve state (*Hayashi et al., 2007*). The same characteristics may be applied to human pluripotent stem cells, although no direct results have yet been shown in human pluripotent stem cells (*Hayashi and Furue, 2016*). IWRLY suspension conditions may allow hiPSCs to adapt easily to form three-dimensional colony morphology by shifting to naïve state.

Last but not least, we successfully generated hiPSC lines from PBMCs in suspension conditions for the first time by adding the abovementioned compounds to the culture medium. Both transduction methods with SeVdp and episomal plasmid vectors enabled the establishment of hiPSCs under IWRLY suspension conditions. Clonal expansion of hiPSCs was also performed by single-cell sorting with flow cytometry, and the established hiPSCs were successfully characterized for their self-renewal, pluripotency, and genomic integrity. These results indicate that hiPSC lines generated in IWRLY suspension conditions have the same quality in terms of their pluripotency, self-renewal, and genomic integrity as those generated with conventional adherent conditions. This consistent process from establishing hiPSCs from somatic cells to their mass expansion with precise control of cellular status in suspension conditions may pave the way for their stable and automated clinical application toward autologous cell therapy of hiPSCs.

# Materials and methods

**Key resources table**

| Reagent type (species) or resource | Designation | Source or reference | Identifiers | Additional information |
|---|---|---|---|---|
| Strain, strain background (*Mus musculus*, male) | NOD.Cg-Prkdcscid Il2rgtm1Wjl/SzJ (NSG) mice, 5 weeks at purchase | The Jackson Laboratory | Strain#: 005557 RRID:IMSR_JAX:05557 | |
| Cell line (*Homo sapiens*) | WTC11 | Coriell Institute; *Hayashi et al., 2016* | GM25256 RRID:CVCL_Y803 | |
| Cell line (*H. sapiens*) | 201B7 | RIKEN Cell Bank; *Takahashi et al., 2007* | HPS0063 RRID:CVCL_A324 | |
| Cell line (*H. sapiens*) | 454E2 | RIKEN Cell Bank; *Okita et al., 2011* | HPS0077 RRID:CVCL_T791 | |
| Cell line (*H. sapiens*) | 1383D6 | RIKEN Cell Bank; *Okita et al., 2011* | HPS1006 RRID:CVCL_UP39 | |
| Cell line (*H. sapiens*) | 1231A3 | RIKEN Cell Bank; *Okita et al., 2011* | HPS0381 RRID:CVCL_LJ39 | |
| Cell line (*H. sapiens*) | Ff-I14s04 | CiRA foundation, Kyoto University; *Kitano et al., 2022* | | |
| Cell line (*H. sapiens*) | HiPS-NB1RGB | RIKEN Cell Bank; *Borisova et al., 2022*; *Shimizu et al., 2022* | HPS5067 | |
| Cell line (*H. sapiens*) | PAX6-TEZ | This paper | | deposited as HPS4903 in RIKEN Cell Bank |
| Cell line (*H. sapiens*) | SOX17-TEZ | This paper | | deposited as HPS4905 in RIKEN Cell Bank |

*Continued on next page*

*Continued*

| Reagent type (species) or resource | Designation | Source or reference | Identifiers | Additional information |
|---|---|---|---|---|
| Biological sample (*H. sapiens*) | Healthy donor-derived PBMCs | Precision for Medicine | Cat#33000-10M | |
| Peptide, recombinant protein | iMatrix-511 silk | Matrixome | Cat#892021 | |
| Peptide, recombinant protein | iMatrix-511MG | Matrixome | Cat#892005 | |
| Peptide, recombinant protein | Vitronectin (VTN-N) Recombinant Human Protein, Truncated | Thermo Fisher Scientific | Cat#A14700 | |
| Peptide, recombinant protein | Accutase | Nacalai Tesque | Cat#12679-54 | |
| Peptide, recombinant protein | TrypLE Select | Thermo Fisher Scientific | Cat#A12859-01 | |
| Peptide, recombinant protein | IL-6 | FUJIFILM Wako Pure Chemical Corporation | Cat#091-07511 | (100 ng/mL) |
| Peptide, recombinant protein | IL-3 | FUJIFILM Wako Pure Chemical Corporation | Cat#092-04621 | (10 ng/mL) |
| Peptide, recombinant protein | SCF | FUJIFILM Wako Pure Chemical Corporation | Cat#195-19071 | (300 ng/mL) |
| Peptide, recombinant protein | TPO | FUJIFILM Wako Pure Chemical Corporation | Cat#200-16471 | (300 ng/mL) |
| Peptide, recombinant protein | FLT3 ligand | FUJIFILM Wako Pure Chemical Corporation | Cat#060-07083 | (300 ng/mL) |
| Peptide, recombinant protein | Activin A | FUJIFILM Wako Pure Chemical Corporation | Cat#014-23961 | (10 ng/mL) |
| Commercial assay or kit | StemFit AK02N medium | Ajinomoto | Cat#AK02N | |
| Commercial assay or kit | StemScale PSC suspension medium | Thermo Fisher Scientific | Cat#A4965001 | |
| Commercial assay or kit | mTeSR1 medium | STEMCELL Technologies | Cat#85850 | |
| Commercial assay or kit | StemFitAK03N | Ajinomoto | Cat#AK03N | |
| Commercial assay or kit | StemSpan-AOF | STEMCELL Technologies | Cat#ST100-0130 | |
| Commercial assay or kit | Human iPS cell Generation Episomal vector Mix | Takara Bio | Cat#3673 | |
| Commercial assay or kit | Amaxa Human CD34+ Cell Nucleofector kit | Lonza | Cat#VPA-1003 | |
| Commercial assay or kit | CytoTune EX-iPS | ID Pharma | Cat#69060-61 | |
| Commercial assay or kit | DMEM high Glucose | Nacalai Tesque | Cat#08458-16 | |
| Commercial assay or kit | 0.1% (w/v) Gelatin Solution | FUJIFILM Wako Pure Chemical Corporation | Cat#190-15805 | |
| Commercial assay or kit | fetal bovine serum | Biosera | Cat#515-99055 | |
| Commercial assay or kit | RPMI1640 medium | FUJIFILM Wako Pure Chemical Corporation | Cat#189-02025 | |
| Commercial assay or kit | B-27 supplement, minus insulin | Thermo Fisher Scientific | A1895601 | |
| Commercial assay or kit | Jes/Wes 12- to 230 kDa separation module for Wes, 8×25 capillary cartridges | ProteinSimple | Cat#SM-W004 | |
| Commercial assay or kit | Anti-Rabbit/Goat/Mouse Detection Module kit | ProteinSimple | Cat#DM-002/DM-001/ DM/006 | |

*Continued on next page*

*Continued*

| Reagent type (species) or resource | Designation | Source or reference | Identifiers | Additional information |
|---|---|---|---|---|
| Commercial assay or kit | FastGene RNA premium kit | NIPPON Genetics | Cat#FG-81250 | |
| Commercial assay or kit | ReverTra Ace qPCR RT kit | TOYOBO | Cat#FSQ-101 | |
| Commercial assay or kit | THUNDERBIRD Probe qPCR Mix | TOYOBO | Cat#QPS-101 | |
| Commercial assay or kit | DNeasy Blood & Tissue Kit | QIAGEN | Cat#69504 | |
| Commercial assay or kit | Karyostat Assay arrays | Thermo Fisher Scientific | Cat#905403 | |
| Chemical compound, drug | Y-27632 | FUJIFILM Wako Pure Chemical Corporation | Cat# HY-10071 | |
| Chemical compound, drug | 0.5 M EDTA solution | Nacalai Tesque | Cat#06894-14 | |
| Chemical compound, drug | Annexin V (Alexa Fluor 680) conjugates | Thermo Fisher Scientific | Cat#A35109 | |
| Chemical compound, drug | DAPI solution | FUJIFILM Wako Pure Chemical Corporation | Cat#340-07971 | |
| Chemical compound, drug | 4% paraformaldehyde in phosphate buffer solution | Nacalai Tesque | Cat#09154-85 | |
| Chemical compound, drug | SB431542 | FUJIFILM Wako Pure Chemical Corporation | Cat#198-16543 | (10 µM) |
| Chemical compound, drug | DMH1 | FUJIFILM Wako Pure Chemical Corporation | Cat#041-33881 | (10 µM) |

## Cell lines

In this study, we used WTC11 (GM25256 from Coriell Institute) (*Hayashi et al., 2016*), 201B7 (HPS0063 from RIKEN Cell Bank) (*Takahashi et al., 2007*), 454E2 (HPS0077 from RIKEN Cell Bank) (*Okita et al., 2011*), 1383D6 (HPS1006 from RIKEN Cell Bank) (*Okita et al., 2011*), 1231A3 (HPS0381 from RIKEN Cell Bank) (*Okita et al., 2011*), Ff-I14s04 (CiRA foundation, Kyoto University) (*Kitano et al., 2022*), and HiPS-NB1RGB (HPS5067 from RIKEN Cell Bank), which were generated from human neonatal skin fibroblast (RCB0222) (*Borisova et al., 2022*; *Shimizu et al., 2022*), as healthy donor hiPSC lines. We generated knock-in hiPSC lines for PAX6-TEZ (constructed from 454E2 line) and SOX17-TEZ (constructed from 1383D6 line) using CRISPR-Cas9 genome editing (*Tsukamoto et al., 2021*; *Figure 1—figure supplement 1A*, Key resources table). We confirmed their identity with STR-PCR profiling and negative for the mycoplasma contamination with PCR method and DNA staining in RIKEN BioResource Research Center (BRC) and CiRA foundation.

## HiPSCs cultured in adherent conditions

These hiPSCs were cultured in StemFit AK02N medium (Cat#AK02N, Ajinomoto). Medium change was performed every day and passaged at 80–90% confluency after 6–7 days of culture. At passage, PBS/EDTA solution (diluted from 0.5 M EDTA solution, Cat#06894-14, Nacalai Tesque, Kyoto, Japan) was used to dissociate hiPSC colonies, and these cells were seeded at a density of 2500 cells/cm$^2$. 10 µM Y-27632 (Cat#HY-10071, FUJIFILM Wako Pure Chemical Corporation, Osaka, Japan) and 0.25 µg/cm$^2$ iMatrix-511 silk (Cat#892021, Matrixome, Osaka, Japan) were added to the culture dish on seeding day. These hiPSC cells were cultured in a $CO_2$ incubator (Forma Steri-Cycle i160, Thermo Fisher Scientific) with a gas conditions at 5% $CO_2$, 21% $O_2$, >95% humidity, and 37°C.

## Suspension culture of hiPSCs

Suspension culture with rotation at 90 rpm was performed with a plate shaker (Cat#WB-101SRC, WAKENBTECH, Kyoto, Japan, or #0081704-000, TAITEC, Tokyo, Japan) installed in a $CO_2$ incubator (Cat#Steri-Cycle i160, Thermo Fisher Scientific) and operated under high humidity continuously during the whole culture period. To start the culture hiPSCs in suspension conditions, $4 \times 10^5$ cells were seeded in one well of a low-attachment 6-well plate (Cat#MS-80060, Sumitomo Bakelite, Tokyo, Japan) with 4 mL of StemFit AK02N medium, StemScale PSC suspension medium (A4965001, Thermo Fisher Scientific), or mTeSR1 medium (Cat#85850, STEMCELL Technologies) supplemented with

10 µM Y-27632. This plate was placed onto the plate shaker in the $CO_2$ incubator (Forma Steri-Cycle i160, Thermo Fisher Scientific). The medium without Y-27632 was changed every day unless otherwise specified. On days 3–5, the hiPSC aggregates were dissociated with Accutase (Cat#12679-54, Nacalai Tesque) at 37°C for 10 minutes. The dissociated cells were counted with an automatic cell counter (Model R1, Olympus) with Trypan Blue staining to detect live/dead cells. These cell suspension was spun down at 200 × $g$ for 3 minutes, and the supernatant was aspirated. The cell pellet was resuspended with a new culture medium at an appropriate cell concentration and used for the next suspension culture. This passage was performed every 5 days unless otherwise specified. To screen for factors that inhibit spontaneous differentiation of hiPSCs, chemicals or recombinant proteins were added to the culture medium (listed in *Supplementary file 1*). These hiPSC cells were cultured in a $CO_2$ incubator with a gas conditions at 5% $CO_2$, 21% $O_2$, >95% humidity, and 37°C.

## Bioreactor culture of hiPSCs

A frozen stock of hiPSCs was pre-cultured twice in adherent conditions to prepare enough cell numbers. To prepare enough hiPSCs to start large-scale culture, hiPSCs were pre-cultured in iMatrix-511MG (Cat#892005, Matrixome) or Vitronectin (VTN-N) Recombinant Human Protein, Truncated (Cat#A14700, Thermo Fisher Scientific)-coated cell culture flasks with StemFitAK03N (Cat#AK03N, Ajinomoto) including 20 µM IWR-1-endo and 1 µM LY333531. A 30 mL stirred suspension bioreactor (BWV-S03A, Able Co., Tokyo, Japan) was used according to a previous study (*Matsumoto et al., 2022*). The medium change was manually performed every other days. After 3–4 days of culture, the formed hiPSC aggregates were dissociated with TrypLE Select (Cat#A12859-01, Thermo Fisher Scientific). As for large-scale hiPSCs culture, the reactor system BioFlo320 (Eppendorf, Hamburg, Germany) was used according to a previous study (*Kropp et al., 2016*). Perfusion culture was started with 3.2–4.8 × $10^7$ cells in 320 mL of StemFit AK03N medium with IWR-1-endo and LY333531. To maintain the lactate concentration below a certain level and regulate the pH, the culture was carried out by increasing the amount of medium perfusion per unit time in accordance with the cell proliferation transition. To prevent pH decrease, $CO_2$ concentration was regulated by feedback control in the reactor system. After 3–4 days of culture, the formed hiPSC aggregates were dissociated with TrypLE Select collected for making cell stocks (~300 tubes). This perfusion culture was repeated three times (P1, P2, and P3) and the cells were prepared at each expansion step.

## HiPSC generation in suspension conditions

HiPSCs were generated from healthy donor-derived PBMCs (Cat#33000-10M, Precision for Medicine). Thawed PMBCs from a vial containing around 1 × $10^7$ cells were pre-cultured in one well of low-attachment 6-well plate including 4 mL of StemSpan-AOF (Cat#ST100-0130, STEMCELL Technologies) supplemented with recombinant human IL-6 (100 ng/mL), IL-3 (10 ng/mL), SCF (300 ng/mL), TPO (300 ng/mL), and FLT3 ligand (300 ng/mL) (all from FUJIFILM Wako Pure Chemical Corporation, Tokyo, Japan). After 24 hours of incubation with continuous stirring at 37°C/5% $CO_2$/21% $O_2$, PBMCs were spun down with centrifugation at 200 × $g$ for 10 minutes at low deceleration speed and resuspended in StemSpan ACF for cell counting. For episomal vector transduction, 2.5 × $10^6$ cells of PMBCs were centrifuged at 200 × $g$ for 10 minutes with low deceleration speed and electroporated using Nucleofector 2b device (Lonza, Basel, Switzerland) with Human iPS cell Generation Episomal vector Mix (Cat#3673, Takara Bio, Shiga, Japan) and Amaxa Human CD34+ Cell Nucleofector kit (Cat#VPA-1003, Lonza) according to the manufacturer's protocol. 5 × $10^5$ of electroporated cells were seeded in one well of a low-attachment 6-well plate in 4 mL StemSpan ACF with the cytokines mentioned above. Suspension culture was performed with continuous agitation at 90 rpm. Stem span ACF medium were gradually replaced with StemFit AK02N medium. Formed cell aggregates were passaged with Accutase on day 16 in the presence of 10 µM Y27632, and suspension culture was continued until the cell numbers reached a sufficient amount for characterization. 10 µM IWR-1-endo and 1 µM LY333531 were added from day 3.

SeVdp infection was performed with Sendai Reprogramming Kit (CytoTune EX-iPS of virus solution, ID Pharma, Tsukuba, Japan) according to the manufacturer's protocol with some modifications. Briefly, pre-cultured 1 × $10^6$ PMBCs were centrifuged at 200 × $g$ for 10 minutes, with low deceleration speed, and resuspended in 2 mL of StemSpan ACF with cytokines. PMBCs were gently mixed with 2 mL of virus solution prepared at MOI = 5 per 1 × $10^6$ cells. 5 × $10^5$ infected cells were seeded in one well of

low-attachment 6-well plate at a total volume of 4 mL with StemSpan ACF plus cytokines. Suspension culture was initiated with continuous agitation at 90 rpm. Stem span ACFs were gradually replaced with StemFit AK02N as mentioned above, and the cell aggregates were passaged with Accutase on day 16 in the presence of 10 µM Y-27632. Suspension culture and passages continued until cell reached a sufficient number for characterization. 10 µM IWR-1-endo and 1 µM LY333531 were added from day 23. These cells were cultured in a $CO_2$ incubator (Forma Steri-Cycle i160, Thermo Fisher Scientific) with a gas conditions at 5% $CO_2$, 21% $O_2$, >95% humidity, and 37°C.

## Three germ layer differentiation in vitro

For EB formation assay, suspension-cultured hiPSC lines were dissociated with Accutase, and 1.0 × $10^4$ cells were seeded in each well of an EZ-BindShut 96-well-V plate (Cat#4420-800SP AGC TECHNO GLASS CO., LTD, Shizuoka, Japan) with 100 µL StemFit AK02N medium supplemented with 10 µM Y-27632. Before culture, the 96-well-V plate was centrifuged at 200 × $g$ for 3 minutes for efficient cell mass formation. The next day, the culture medium was switched to DMEM high Glucose (Cat#08458-16, Nacalai Tesque) supplemented with 10% fetal bovine serum (Cat#515-99055, Biosera; hereafter referred to as EB medium). On day 8, EBs were transferred into 0.1% (w/v) Gelatin Solution (Cat#190-15805, FUJIFILM Wako Pure Chemical Corporation)-coated 12-well plate and further cultured in EB medium for another 8 days. The medium was changed every day. EBs were fixed with 4% paraformaldehyde in phosphate buffer solution (Cat#09154-85, Nacalai Tesque) for 10 minutes at room temperature and used for immunostaining. The differentiation was validated by immunostaining against each germ layer markers.

The differentiation potency of suspension-cultured hiPSCs toward three germ layers was also evaluated by culturing in germ layer-specific differentiation medium. As in the maintenance conditions, 4 × $10^5$ hiPSCs were seeded in one well of a low-attachment 6-well plate with 4 mL of StemFit AK02N medium supplemented with 10 µM Y-27632. This plate was placed onto the plate shaker in the $CO_2$ incubator. Next day, the medium was changed to the germ layer-specific differentiation medium. For ectodermal differentiation, suspension-cultured hiPSCs spheroids were cultured with StemFit AK02N without C medium supplemented with 10 µM SB431542 (Cat#198-16543, FUJI-FILM Wako Pure Chemical Corporation) and 10 µM DMH1 (Cat#041-33881, FUJIFILM Wako Pure Chemical Corporation) for 7 days. For mesodermal differentiation, suspension-cultured hiPSCs spheroids were cultured with RPMI1640 medium (Cat#189-02025, FUJIFILM Wako Pure Chemical Corporation) supplemented with B-27 supplement, minus insulin (A1895601, Thermo Fisher Scientific, MO) and 7.5 µM CHIR99021 (Cat#034-23103, FUJIFILM Wako Pure Chemical Corporation) for 1 day, and continuously with RPMI1640 medium supplemented with B-27 supplement for 2 days. For endodermal differentiation, suspension-cultured hiPSCs spheroids were cultured with RPMI1640 medium supplemented with 1 mM sodium pyruvate (FUJIFILM Wako Pure Chemical Corporation), 1× NEAA (FUJIFILM Wako Pure Chemical Corporation), 80 ng/mL Activin A (R&D Systems, MN), 55 µM 2-mercaptethanol (FUJIFILM Wako Pure Chemical Corporation), 50 ng/mL FGF2 (R&D Systems), 20 ng/mL BMP4, and 3 µM CHIR99021 for 2 days, and continuously with RPMI1640 medium supplemented with 1 mM sodium pyruvate, 1× NEAA, 80 ng/mL Activin A, 55 µM 2-mercaptethanol, and 0.5% knockout serum replacement (KSR; Thermo Fisher Scientific) for 2 days.

## Neuroectoderm and endoderm differentiation of reporter hiPSC lines

Reporter hiPSC lines, PAX6-TEZ, and SOX17-TEZ were used as positive controls for in vitro neuroectoderm and endoderm differentiation, respectively. PAX6-TEZ was seeded in each well of a 24-well plate coated with 0.25 µg/cm² iMatrix-511 silk with 1 mL StemFit AK02N medium supplemented with 10 µM Y-27632. Next day, the culture medium was switched to StemFit AK02N medium without supplement C, instead of containing 10 µM SB431542 and 10 µM DMH1. SOX17-TEZ was seeded in each well of a 24-well plate (same as above), and on the next day, the culture medium was switched to StemFit AK02N medium without supplement C instead of containing 3 µM CHIR99021 and 10 ng/mL Activin A (Cat#014-23961, FUJIFILM Wako Pure Chemical Corporation). The medium was changed every day. On day 7, tdTomato expression was observed under all-in-one fluorescent microscope (BZ-X800; KEYENCE, Osaka, Japan).

## Cardiomyocyte differentiation

Cardiomyocyte differentiation was performed according to a modified method described previously (*Funakoshi et al., 2016*). Both large-scale suspension-cultured hiPSCs and typical adherent-cultured hiPSCs were collected by centrifugation after dissociation into single cell with TrypLE Select. The cells were suspended in 1.5 mL cardiomyocyte differentiation media (CDM), consisting of StemPro34 medium (Thermo Fisher Scientific) supplemented with 2 mM GlutaMAX (Thermo Fisher Scientific), 50 µg/mL ascorbic acid, $4 \times 10^{-4}$ M monothioglycerol (Sigma-Aldrich), 150 µg/mL transferrin (Roche), and with Matrigel (Cat#354277, Corning, NY, USA), 10 µM Y-27632 and 2 ng/mL human recombinant BMP4 (R&D Systems), and then cultured in ultra-low-attachment 6-well plate (Corning) in 5% $CO_2$/5% $O_2$. After 24 hours, 1.5 mL CDM with 6 ng/mL human recombinant activin A (R&D Systems) and 5 ng/mL bFGF (R&D Systems) were added into the wells. After 3 days, medium was changed to 3 mL CDM with 10 ng/mL VEGF (R&D Systems), SB431542 (Sigma-Aldrich), dorsomorphin (Sigma-Aldrich), and 1 µM IWP-3 (Stemgent). After 7 days, medium was changed to 3 mL CDM with 10 ng/mL VEGF (R&D Systems). The medium was changed every 2 days. The cells were collected at 15 days after differentiation and analyzed by flow cytometry.

## Dopaminergic progenitor cells differentiation

Dopaminergic progenitor cells differentiation was performed according to a modified method described previously (*Doi et al., 2020*). Both large-scale suspension-cultured hiPSCs and typical adherent-cultured hiPSC were collected by centrifugation after dissociation into single cell with TrypLE Select. The cells were suspended in 1 mL dopaminergic progenitor cells differentiation medium-1 (DPM-1), consisting of Glasgow's minimum essential medium (Thermo Fisher) supplemented with 8% knockout serum replacement (Thermo Fisher Scientific), 1% MEM Non-Essential Amino Acids Solution (Thermo Fisher Scientific), 1 mM sodium pyruvate (Sigma-Aldrich), 0.1 mM 2-mercaptoethanol (FujiFilm Wako), and 100 nM LDN193189 (Stemgent), and were then seeded in iMatrix-coated 24-well plates at a density of $1 \times 10^6$ cells/well with 10 µM Y-27632 and 500 nM A-83-01 (FujiFilm Wako). After 1 and 2 days, medium was changed to DPM-1 with 500 nM A-83-01, 100 ng/mL recombinant human FGF8 (FujiFilm Wako), and 2 µM purmorphamine (FujiFilm Wako). After 3–6 days, medium was changed every day to DPM-1 with 500 nM A-83-01, 100 ng/mL recombinant human FGF8, 2 µM purmorphamine, and 3 µM CHIR99021 (FujiFilm Wako). After 7–11 days, medium was changed every day to DPM-1 with 3 µM CHIR99021. On differentiation day 12, cells were dissociated using TrypLE Select and suspended in the dopaminergic progenitor cells differentiation medium-2 (DPM-2), consisting Neurobasal medium (Thermo Fisher Scientific), 2% B27 supplement (without vitamin A, Thermo Fisher), 1% GlutaMAX (Thermo Fisher Scientific), 10 ng/mL human recombinant glial cell-derived neurotrophic factor (FujiFilm Wako), 200 mM ascorbic acid, 20 ng/mL human recombinant brain-derived neurotrophic factor (FujiFilm Wako), and 400 µM dibutyryl cAMP (FujiFilm Wako), and were then plated on U-shaped 96-well plates (Thermo Fisher Scientific) at a density of $2 \times 10^4$ cells/150 µL/well with 10 µM Y-27632. After 15–26 days, half of the medium was changed every 2 days to DPM-2. Dopaminergic progenitor cells differentiation efficiency was analyzed by flow cytometry at 12 days after differentiation. On differentiation day 12, cells were labeled with anti-CORIN antibody (Clone 5B6, Sigma-Aldrich) and secondary antibody (A11001, Thermo Fisher).

## Hepatocyte differentiation

Hepatocyte differentiation was performed according to a modified method described previously (*Si Tayeb et al., 2010*). Both large-scale suspension-cultured and typical adherent-cultured hiPSCs were collected by centrifugation after dissociation into single cell with TrypLE Select. These hiPSCs were cultured on Matrigel-coated 6-well plates until 80% confluency with StemFit medium (Ajinomoto). To initiate differentiation, the medium was changed to 1 mL hepatocyte differentiation medium (HDM) consisting of RPMI1640 medium (Thermo Fisher) supplemented with 1×GlutaMAX (Thermo Fisher) and 1×B27 supplement minus vitamin A, with the addition of 100 ng/mL Activin A (R&D Systems). After 1–3 days, medium was changed every day to HDM with 100 ng/mL Activin A. After 4–8 days, medium was changed every day to HDM with 20 ng/mL human recombinant BMP-4 (R&D Systems) and 20 ng/mL human recombinant FGF-4 (R&D Systems). After 9–13 days, medium was changed every day to HDM with 20 ng/mL human recombinant HGF (R&D Systems). After 14–24 days, medium was changed every 2 days to Hepatocyte Culture Medium (Lonza) with 20 ng/mL human recombinant

Oncostatin M (R&D Systems). On differentiation day 25, the albumin concentration in the culture supernatant was determined by ELISA using an anti-human albumin antibody (Betyl Laboratories, Cat#E88-129).

## Teratoma formation

Suspension-cultured hiPSCs were dissociated with Accutase and then resuspended with 1 mL of StemFit AK02N medium supplemented with 10 µM Y-27632. $1 \times 10^6$ cells were collected by centrifugation at $200 \times g$ for 3 minutes and suspended in ice-cold 50% Matrigel solution: StemFit AK02N with 10 µM Y-27632=1:1. Cells were injected into testis or leg of NOD.Cg-Prkdcscid Il2rgtm1Wjl/SzJ (NSG) mice (The Jackson Laboratory, Bar Harbor, ME) using an 18-G needle. Two to three months later, teratomas were collected and fixed with 4% paraformaldehyde in D-PBS. Paraffin-embedded sectioning and hematoxylin-eosin (HE) staining was performed in Genostaff Inc, Tokyo, Japan. Three germ layer derivatives were observed under CKX53 microscope with a DP22 camera and CellSens software (EVIDENT, Tokyo, Japan).

## Apoptosis assay of suspension-cultured iPSCs

Suspension-cultured hiPSCs were dissociated with Accutase. The dissociated cells were aliquoted into $1 \times 10^5$ cells/100 µL with a binding buffer consisting of 10 mM HEPES, 140 mM NaCl, 2.5 mM $CaCl_2$, then added with 5 µL Annexin V (Alexa Fluor 680) conjugates (Cat#A35109, Thermo Fisher Scientific) and 1 µL DAPI solution (Cat#340-07971, FUJIFILM Wako Pure Chemical Corporation). After incubation for 15 minutes at room temperature in dark conditions, 400 µL of cold binding buffer was added. The apoptotic cells were immediately analyzed with a flow cytometer (SH800S, SONY, Tokyo, Japan). As a positive control of apoptosis, suspension-cultured hiPSCs were treated with Staurosporine (1 µM) (Cat#S1421, Selleck Biotech, Tokyo, Japan), one of the apoptosis-inducing factors, for 2 hours before collection.

## Karyotyping

For virtual karyotyping, genomic DNA (gDNA) was extracted from hiPSCs using a DNeasy Blood & Tissue Kit (Cat#69504, QIAGEN, Hulsterweg, the Netherlands) and was used for microarray assay. Virtual karyotyping was performed with GeneChip Scanner System 3000 using Karyostat Assay arrays (Cat#905403, both from Thermo Fisher Scientific) according to the manufacturer's protocol. Data were analyzed using the Chromosome Analysis Suite (ChAS) and Affymetrix GeneChip Command Console software programs.

G-band analysis was performed using the common Giemsa staining method with hiPSCs fixed by Carnoy's fixative (3:1 ratio of methanol:glacial acetic acid).

## qRT-PCR

Total RNA was extracted with a FastGene RNA premium kit (Cat#FG-81250, NIPPON Genetics, Tokyo, Japan) and used for reverse transcription reaction. cDNA was synthesized by using a ReverTra Ace qPCR RT kit (Cat#FSQ-101, TOYOBO, Osaka, Japan) with random primers. Real-Time qPCR reactions were performed with a QuantStudio 3 System (Thermo Fisher Scientific) using THUNDERBIRD Probe qPCR Mix (Cat#QPS-101, TOYOBO) with TaqMan probes (listed in *Supplementary file 1*) (Thermo Fisher Scientific) according to the manufacturer's instructions. Gene expression was described as the fold change relative to the control sample value (ΔΔCt method) after normalization to the corresponding GAPDH or β-Actin values, unless otherwise specified. For the residual SeV detection, data were normalized to the expression of GAPDH and displayed as a relative fold increase to hiPSC line established with episomal vector (WTC11 line). SeVdp-infected fibroblasts on passage 1 (SeV-34 and –39) were used as positive controls.

## Immunocytochemistry

Immunocytochemistry was performed on adherent cells. Suspension-cultured hiPSCs were dissociated with Accutase and transiently cultured as adherent in iMatrix-coated culture dishes for 3 hours before immunocytochemistry. The cells were fixed with 4% paraformaldehyde in D-PBS for 10 minutes at room temperature, then permeabilized in D-PBS containing 0.1% Triton X-100 for another 10 minutes. Cells were incubated with primary antibodies in D-PBS containing 0.1% bovine serum albumin (BSA;

Cat#017-22231, FUJIFILM Wako Pure Chemical Corporation) overnight at 4°C. The secondary antibodies were incubated for 1 hour at room temperature in D-PBS containing 0.1% BSA. Fluoro-KEEPER Antifade Reagent, Non-Hardening Type with DAPI (Cat#12745-74, Nacalai Tesque) was used for nuclear counterstaining. Fluorescence images were taken with an all-in-one fluorescent microscope (BZ-X800; KEYENCE). The primary and secondary antibodies used in this study are listed in *Supplementary file 3*, and *Supplementary file 4*, respectively.

## Automatic capillary western blot (Simple Western assays)

To detect the expression of PAX6 or SOX17, adherent- or suspension-cultured PAX6-TEZ or SOX17-TEZ hiPSCs were collected on day 10 (passage 2). The cell lysate was prepared from 1 × 10⁶ cells with sodium dodecyl sulfate-polyacrylamide gel electrophoresis sample buffer solution without 2-ME (2×) (Cat#30567-12, Nacalai Tesque) supplemented with 100 mM dithiothreitol (Cat#14130-41, Nacalai Tesque). To detect phosphorylated PKCβ, cell lysates were collected on day 5 of suspension-cultured hiPSCs with extraction buffer mentioned above, containing 1% phosphatase inhibitor cocktail (Cat#07574-61, Nacalai Tesque). Samples were denatured at 95°C for 5 minutes. Western blotting was performed with a capillary automatic western blotting device (Simple Western, Wes; Bio-Techne, CA). Jes/Wes 12–230 kDa separation module for Wes, 8×25 capillary cartridges (Cat#SM-W004, ProteinSimple), and Anti-Rabbit/Goat/Mouse Detection Module kit (Cat#DM-002/DM-001/DM/006, ProteinSimple) were used. Preparation of reagents and sample loading was done according to the manufacturer's instructions. The data were analyzed and quantified by compass for Simple Western software program (ProteinSimple). GAPDH was used as the reference for normalization. The primary antibodies used in this study are listed in *Supplementary file 3*.

## Flow cytometry

Flow cytometry of self-renewal markers, TRA-1-60, and SSEA4 was performed as described in our previous study. Briefly, adherent- or suspension-cultured hiPSCs were dissociated with 0.5 mM EDTA solution in D-PBS or Accutase, respectively. Dissociated cells (0.5 or 1 × 10⁶ cells) were collected and centrifuged at 200 × *g* for 3 minutes and resuspended in 500 or 1000 μL PBS containing 0.1% BSA and 0.5 mM EDTA, then incubated with or without anti-TRA-60 antibody or anti-SSEA4 antibody for 1 hour at 4°C under oblique light conditions. After washing with PBS, cells were resuspended in 500 μL D-PBS containing 0.1% BSA and 0.5 mM EDTA, and filtrated with 35 μm cell strainer (Cat#352235, FALCON, Thermo Fisher Scientific). For PAX6-TEZ and SOX17-TEZ hiPSC lines, adherent- or suspension-cultured hiPSCs were dissociated into single cells and collected cells were resuspended in 500 μL PBS containing 0.1% BSA and 0.5 mM EDTA. After passing through the 35 μm cell strainer, cells were analyzed for tdTomato. The primary and secondary antibodies used in this study are listed in *Supplementary file 3* and *Supplementary file 4*, respectively. Flow cytometry was performed with SH800 cell Sorter (SONY).

## RNA-seq

Total RNA was extracted using the FastGene RNA premium kit, and strand-specific library preparation was performed. The prepared library was sequenced using a NovaSeq6000 (Illumina, Inc, CA). Sequencing was performed in a 150 bp ×2 paired-end configuration with a data output of about 6 Gb per sample (~20 million paired reads). Library preparation and sequencing were performed in GENEWIZ (Azenta, MA). To identify differentially regulated genes, sequencing data was analyzed with a CLC Genomics Workbench (QIAGEN) and R package, edgeR (v3.30.3) in R language (v 4.1.0). An MA plot (log2 fold change versus mean average expression) comparing transcriptomes between suspension and adherent conditions from RNA-seq data. Transcripts with log2 fold change ≧ 2 or ≦ –2 (false discovery rate [FDR] < 0.01) are highlighted with red and blue dots, respectively. The extracted genes were analyzed for Gene Ontology enrichment and biological pathways using the R package, clusterProfiler (v3.16.1). GSEA was also performed using clusterProfiler. An enrichment map was used to visualize the GSEA results with enrichment score (ES) and FDR values. The software used for the analysis is listed in *Supplementary file 5*.

## Statistical analysis

Statistical analysis was performed using Student's *t*-test, Dunnett's test for multiple comparisons with a single control condition, and one-way ANOVA and Tukey's tests for multiple comparisons with all the conditions. p-values <0.05 were considered statistically significant. *, **, or *** in the graphs indicate p<0.05, p<0.01, or p<0.001, respectively. No statistical methods were used to predetermine sample size. The experiments were not randomized and the investigators were not blinded to allocation during experiments and outcome assessment.

## Acknowledgements

We thank Dr. Bruce Conklin for providing us with the WTC11 hiPSC line; Dr. Shinya Yamanaka, Dr. Kazutoshi Takahashi, Dr. Masato Nakagawa, and Dr. Keisuke Okita for providing us with 201B7, 454E2, 1231A3, 1383D6, and Ff-I14s04 hiPSC lines. We would like to express our sincere gratitude to Dr. Kaoru Saijo, Mr. Daiki Kondo, and Mr. Masahiko Yamada for their technical assistance, and Ms. Kumiko Omori for her administrative support. This work was funded by a grant from AMED (23bm1423010h0001) to Y.Ha., a collaborative research grant from Kaneka corporation to Y.Ha., and an endowment to CiRA foundation. We would like to thank Editage (http://www.editage.com) for editing and reviewing this manuscript for English language.

## Additional information

### Competing interests

Mami Matsuo-Takasaki, Yohei Hayashi: Inventor of patents arising from this work (WO/2021/162090 and PCT/JP2020/005255). Sho Kambayashi, Kazuhiro Takeuchi, Masato Ibuki: Was employed with KANEKA corporation during this study. Inventor of patents arising from this work (WO/2021/162090 and PCT/JP2020/005255). Terasu Kawashima, Rio Masayasu, Manami Suzuki, Yoshikazu Kawai, Tomoyuki Nakaishi, Naoki Nishishita: Was employed with KANEKA corporation during this study. The other authors declare that no competing interests exist.

### Funding

| Funder | Grant reference number | Author |
|---|---|---|
| Japan Agency for Medical Research and Development | JP23bm1423010 | Yohei Hayashi |
| Japan Agency for Medical Research and Development | JP15bm0104001 | Masayoshi Tsukahara |
| Kaneka Corporation | | Yohei Hayashi |
| CiRA Foundation | | Yohei Hayashi |

The funders had no role in study design, data collection and interpretation, or the decision to submit the work for publication.

### Author contributions

Mami Matsuo-Takasaki, Resources, Data curation, Formal analysis, Validation, Investigation, Visualization, Methodology, Writing – original draft, Project administration, Writing – review and editing; Sho Kambayashi, Data curation, Formal analysis, Validation, Investigation, Visualization, Methodology, Writing – original draft, Writing – review and editing; Yasuko Hemmi, Tomoya Shimizu, Manami Suzuki, Validation, Investigation, Methodology, Writing – review and editing; Tamami Wakabayashi, Yuri An, Hidenori Ito, Terasu Kawashima, Rio Masayasu, Investigation, Visualization, Methodology, Writing – review and editing; Kazuhiro Takeuchi, Formal analysis, Investigation, Visualization, Methodology, Writing – review and editing; Masato Ibuki, Validation, Investigation, Visualization, Methodology, Project administration, Writing – review and editing; Yoshikazu Kawai, Validation, Investigation, Visualization, Project administration, Writing – review and editing; Masafumi Umekage, Tomoaki M

Kato, Formal analysis, Investigation, Visualization, Writing – review and editing; Michiya Noguchi, Koji Nakade, Resources, Validation, Investigation, Visualization, Writing – review and editing; Yukio Nakamura, Resources, Supervision, Validation, Investigation, Project administration, Writing – review and editing; Tomoyuki Nakaishi, Validation, Investigation, Project administration, Writing – review and editing; Naoki Nishishita, Supervision, Validation, Investigation, Methodology, Project administration, Writing – review and editing; Masayoshi Tsukahara, Supervision, Funding acquisition, Validation, Investigation, Visualization, Project administration, Writing – review and editing; Yohei Hayashi, Conceptualization, Resources, Data curation, Formal analysis, Supervision, Funding acquisition, Validation, Investigation, Visualization, Methodology, Writing – original draft, Project administration, Writing – review and editing

### Author ORCIDs
Yohei Hayashi  https://orcid.org/0000-0001-5490-7052

### Ethics

The generation and use of hiPSCs were approved by the Ethics Committee of RIKEN, CiRA Foundation, and KANEKA CORPORATION (Approval numbers: RIKEN-T-2023-004 in RIKEN, G1432 in CiRA Foundation, and 2024-9 in KANEKA CORPORATION). We only used existing cell lines or commercial human materials, which were not necessary with informed consent.

All animal experiments were approved by the Animal Experimentation Committee of the RIKEN Tsukuba branch (Approval number: T-2023-007) and were performed according to the committee's guiding principles and the "Guide for the Care and Use of Laboratory Animals" published by the National Institute of Health.

Reviewer #1 (Public review): https://doi.org/10.7554/eLife.89724.3.sa1
Reviewer #2 (Public review): https://doi.org/10.7554/eLife.89724.3.sa2
Reviewer #3 (Public review): https://doi.org/10.7554/eLife.89724.3.sa3
Author response https://doi.org/10.7554/eLife.89724.3.sa4

## Additional files

### Supplementary files
- Supplementary file 1. The list of chemicals and cytokines used in the screening assay.
- Supplementary file 2. The list of TaqMan probes and primer sets used in this study.
- Supplementary file 3. The list of primary antibodies used in this study.
- Supplementary file 4. The list of secondary antibodies used in this study.
- Supplementary file 5. Software used in this study.
- MDAR checklist

### Data availability

Sequencing data have been deposited in GEO under accession code GSE222833. This paper does not report original codes. The authors declare that all other data supporting the findings of this study are available within the paper and its supplementary files.

The following dataset was generated:

| Author(s) | Year | Dataset title | Dataset URL | Database and Identifier |
|---|---|---|---|---|
| Hayashi Y | 2023 | The effect of signal inhibitors on human induced pluripotent stem cells in suspension culture conditions | https://www.ncbi.nlm.nih.gov/geo/query/acc.cgi?acc=GSE222833 | NCBI Gene Expression Omnibus, GSE222833 |

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
