## [Editor Report · eLife Assessment]

This comprehensive and **compelling** study presents a robust, cost-effective method for expanding pluripotent stem cells. The authors have identified a media condition that maintains iPSCs in suspension cultures by inhibiting the PKCβ and Wnt signaling pathways. The article is **important** for the pluripotent stem cell field as it seeks robust and economical approaches to expand iPSCs at scale for high-throughput screens and preclinical studies. While the authors have tested their media and protocol on a few lines, given the variability of iPSCs, further testing across more cell lines and in different laboratory settings will be crucial to evaluate its reproducibility.

---

## [Referee Report · Reviewer #1 (Public review)]

Summary:

The authors have presented data showing that there is a greater amount of spontaneous differentiation in human pluripotent cells cultured in suspension vs static and have used PKCβ and Wnt signaling pathway inhibitors to decrease the amount of differentiation in suspension culture.

Strengths:

This is a very comprehensive study that uses a number of different rector designs and scales in addition to a number of unbiased outcomes to determine how suspension impacts the behaviour of the cells and in turn how the addition of inhibitors counteracts this effect. Furthermore, the authors were also able to derive new hiPSC lines in suspension with this adapted protocol.

Weaknesses:

The main weakness of this study is the lack of optimization with each bioreactor change. It has been shown multiple times in the literature that the expansion and behaviour of pluripotent cells can be dramatically impacted by impeller shape, RPM, reactor design and multiple other factors. It remains unclear to me how much of the results the authors observed (e.g. increased spontaneous differentiation) was due to not having an optimized bioreactor protocol in place (per bioreactor vessel type). For instance - was the starting seeding density, RPM, impeller shape, feeding schedule, and/or anything other aspect optimized for any of the reactors used in the study and if not, how were the values used in the study determined?

Post-revision:

The authors did a commendable job in responding and addressing my comments and concerns in addition to those of the other reviewers. I think this study will be of interest to the field and will add to our collective knowledge on how PSCs react to being cultured in suspension conditions.

---

## [Referee Report · Reviewer #2 (Public review)]

This study by Matsuo-Takasaki et al. reported the development of a novel suspension culture system for hiPSC maintenance using Wnt/PKC inhibitors. The authors showed elegantly that inhibition of the Wnt and PKC signaling pathways would repress spontaneous differentiation into neuroectoderm and mesendoderm in hiPSCs, thereby maintaining cell pluripotency in suspension culture. This is a solid study with substantial data to demonstrate the quality of the hiPSC maintained in the suspension culture system, including long-term maintenance in >10 passages, robust effect in multiple hiPSC lines, and a panel of conventional hiPSC QC assays. Notably, large-scale expansion of a clinical grade hiPSC using a bioreactor was also demonstrated, which highlighted the translational value of the findings here. In addition, the author demonstrated a wide range of applications for the IWR1+LY suspension culture system, including support for freezing/thawing and PBMC-iPSC generation in suspension culture format. The novel suspension culture system reported here is exciting, with significant implications in simplifying the current culture method of iPSC and upscaling iPSC manufacturing.

Review for second submission:

In this revised manuscript, the authors provided new data to further support that suspension culture with Wnt/PKC inhibitors can be used for long-term hiPSC maintenance across multiple cell lines, as well as comparison with current benchmark culture system. New discussion sections were also added to put the findings into perspective of current development and the need for hiPSC maintenance culture system, and the figures were updated to improve readability. Overall, the authors have addressed all my concerns in this revised manuscript. Congratulations to the authors on this very interesting study.

---

## [Referee Report · Reviewer #3 (Public review)]

In the current manuscript, Matsuo-Takasaki et al. demonstrate that the addition of PKCβ and WNT signaling pathway inhibitors to suspension cultures of iPSCs effectively suppresses spontaneous differentiation. These conditions are well-suited for the large-scale expansion of iPSCs. The authors have shown that, under these conditions, they can successfully perform single-cell cloning, direct cryopreservation, and iPSC derivation from PBMCs. Furthermore, they provide a comprehensive characterization of iPSCs grown in these conditions, including assessments of undifferentiated stem cell markers and genetic stability.

They have elegantly demonstrated that iPSCs cultured in these conditions can differentiate into derivatives of all three germ layers. By differentiating iPSCs into dopaminergic neural progenitors, cardiomyocytes, and hepatocytes, the authors show that differentiation is comparable to that of adherent cultures. This new method of expanding iPSCs has significant potential for clinical applications. The authors also tested these conditions in multiple cell lines and observed consistent results.

Although the authors have elaborated on the mechanism to some extent-suggesting that PKCβ and WNT signaling pathway inhibition suppresses differentiation and shifts cells toward a naïve pluripotency state in suspension cultures-further research is needed to fully understand this process. Nevertheless, their findings are promising and will be beneficial for producing scalable amounts of iPSCs in controlled conditions.

---

## [Author Response]

The following is the authors’ response to the previous reviews.

**Public Reviews:**

**Reviewer #1 (Public Review):**
Summary:The authors have presented data showing that there is a greater amount of spontaneous differentiation in human pluripotent cells cultured in suspension vs static and have used PKCβ and Wnt signaling pathway inhibitors to decrease the amount of differentiation in suspension culture.Strengths:This is a very comprehensive study that uses a number of different rector designs and scales in addition to a number of unbiased outcomes to determine how suspension impacts the behaviour of the cells and in turn how the addition of inhibitors counteracts this effect. Furthermore, the authors were also able to derive new hiPSC lines in suspension with this adapted protocol.Weaknesses:The main weakness of this study is the lack of optimization with each bioreactor change. It has been shown multiple times in the literature that the expansion and behaviour of pluripotent cells can be dramatically impacted by impeller shape, RPM, reactor design, and multiple other factors. It remains unclear to me how much of the results the authors observed (e.g. increased spontaneous differentiation) was due to not having an optimized bioreactor protocol in place (per bioreactor vessel type). For instance - was the starting seeding density, RPM, impeller shape, feeding schedule, and/or any other aspect optimized for any of the reactors used in the study, and if not, how were the values used in the study determined?

Thank you for your thoughtful comments. According to your comments, we have performed several experiments to optimize the bioreactor conditions in revised manuscripts. We tested several cell seeding densities and several stirring speeds with or without WNT/PKCβ inhibitors (Figure 6—figure supplement 1). We found that 1 - 2 x 105 cells/mL of the seeding densities and 50 - 150 rpm of the stirring speeds were applicable in the proliferation of these cells. Also, PKCβ and Wnt inhibitors suppressed spontaneous differentiation in bioreactor conditions regardless with stirring speeds. As for the impeller shape and reactor design, we just used commonly-used ABLE's bioreactor for 30 mL scale and Eppendorf's bioreactors for 320 mL scale, which had been designed and used for human pluripotent stem cell culture conditions in previous studies, respectively (Matsumoto et al., 2022 (doi: 10.3390/bioengineering9110613); Kropp et al., 2016 (doi: 10.5966/sctm.2015-0253)). We cited these previous studies in the Results and Materials and Methods section. We believe that these additional data and explanation are sufficient to satisfy your concerns on the optimization of bioreactor experiments.

**Reviewer #2 (Public Review):**
This study by Matsuo-Takasaki et al. reported the development of a novel suspension culture system for hiPSC maintenance using Wnt/PKC inhibitors. The authors showed elegantly that inhibition of the Wnt and PKC signaling pathways would repress spontaneous differentiation into neuroectoderm and mesendoderm in hiPSCs, thereby maintaining cell pluripotency in suspension culture. This is a solid study with substantial data to demonstrate the quality of the hiPSC maintained in the suspension culture system, including long-term maintenance in >10 passages, robust effect in multiple hiPSC lines, and a panel of conventional hiPSC QC assays. Notably, large-scale expansion of a clinical grade hiPSC using a bioreactor was also demonstrated, which highlighted the translational value of the findings here. In addition, the author demonstrated a wide range of applications for the IWR1+LY suspension culture system, including support for freezing/thawing and PBMC-iPSC generation in suspension culture format. The novel suspension culture system reported here is exciting, with significant implications in simplifying the current culture method of iPSC and upscaling iPSC manufacturing.Another potential advantage that perhaps wasn't well discussed in the manuscript is the reported suspension culture system does not require additional ECM to provide biophysical support for iPSC, which differentiates from previous studies using hydrogel and this should further simplify the hiPSC culture protocol.Interestingly, although several hiPSC suspension media are currently available commercially, the content of these suspension media remained proprietary, as such the signaling that represses differentiation/maintains pluripotency in hiPSC suspension culture remained unclear. This study provided clear evidence that inhibition of the Wnt/PKC pathways is critical to repress spontaneous differentiation in hiPSC suspension culture.I have several concerns that the authors should address, in particular, it is important to benchmark the reported suspension system with the current conventional culture system (eg adherent feeder-free culture), which will be important to evaluate the usefulness of the reported suspension system.

Thank you for this insightful suggestion. In this revised manuscript, we have performed additional experiments using conventional media, mTeSR1 (Stem Cell Technologies, Vancouver, Canada), comparing with the adherent feeder-free culture system in four different hiPSC lines simultaneously. Compared to the adherent conditions, the suspension conditions without chemical treatment decreased the expression of self-renewal marker genes/proteins and increased the expression levels of *SOX17*, *T*, and *PAX6* (Figure 4 - figure supplement 2). Importantly, the treatment of LY333531 and IWR-1-endo in mTeSR1 medium reversed the decreased expression of these undifferentiated markers and suppressed the increased expression of differentiation markers in suspension culture conditions, reaching the comparable levels of the adherent culture conditions. These results indicated that these chemical treatments in suspension culture are beneficial even when using a conventional culture medium.

Also, the manuscript lacks a clear description of a consistent robust effect in hiPSC maintenance across multiple cell lines.

Thank you for this insightful suggestion. We have performed additional experiments on hiPSC maintenance across 5 hiPSC lines in suspension culture using StemFit AK02N medium simultaneously (Figure 3C - E). Overall, the treatment of LY333531 and IWR-1-endo in the StemFit AK02N medium reversed the decreased expression of these undifferentiated markers and suppressed the increased expression of differentiation markers in suspension culture conditions. Also as above, we have added results using conventional media, mTeSR1, in comparison to the adherent feeder-free culture system in four different hiPSC lines simultaneously. These results show that this chemical treatment consistently produced robust effects in hiPSC maintenance across multiple cell lines using multiple conventional media.

There are also several minor comments that should be addressed to improve readability, including some modifications to the wording to better reflect the results and conclusions.

In the revised manuscript, we have added and corrected the descriptions to improve readability, including some modifications to the wording to better reflect the results and conclusions.

**Reviewer #3 (Public Review):**
In the current manuscript, Matsuo-Takasaki et al. have demonstrated that the addition of PKCβ and WNT signaling pathway inhibitors to the suspension cultures of iPSCs suppresses spontaneous differentiation. These conditions are suitable for large-scale expansion of iPSCs. The authors have shown that they can perform single-cell cloning, direct cryopreservation, and iPSC derivation from PBMCs in these conditions. Moreover, the authors have performed a thorough characterization of iPSCs cultured in these conditions, including an assessment of undifferentiated stem cell markers and genetic stability. The authors have elegantly shown that iPSCs cultured in these conditions can be differentiated into derivatives of three germ layers. By differentiating iPSCs into dopaminergic neural progenitors, cardiomyocytes, and hepatocytes they have shown that differentiation is comparable to adherent cultures.This new method of expanding iPSCs will benefit the clinical applications of iPSCs.Recently, multiple protocols have been optimized for culturing human pluripotent stem cells in suspension conditions and their expansion. Additionally, a variety of commercially available media for suspension cultures are also accessible. However, the authors have not adequately justified why their conditions are superior to previously published protocols (indicated in Table 1) and commercially available media. They have not conducted direct comparisons.

Thank you for this careful suggestion. In this revised manuscript, we have added results using a conventional medium, mTeSR1 (Stem Cell Technologies), which has been used for the suspension culture in several studies. Compared to the adherent conditions using mTeSR1 medium, the suspension conditions with the same medium decreased the ratio of TRA1-60/SSEA4-positive cells and OCT4positive cells and the expression levels of OCT4 and NANOG and decreased the expression levels of *SOX17*, *T*, and *PAX6* in 4 different hiPSC lines simultaneously (Figure 4 - Supplement 2). Importantly, the treatment of LY333531 and IWR-1-endo in the mTeSR1 medium reversed the decreased expression of these undifferentiated markers. With these direct comparisons, we were able to justify why our conditions are superior to previously published protocols using commercially available media.

Additionally, the authors have not adequately addressed the observed variability among iPSC lines. While they claim in the Materials and Methods section to have tested multiple pluripotent stem cell lines, they do not clarify in the Results section which line they used for specific experiments and the rationale behind their choices. There is a lack of comparison among the different cell lines. It would also be beneficial to include testing with human embryonic stem cell lines.

Thank you for this insightful suggestion. In this revised manuscript, we have added results on 5 different hiPSC lines at the same time (Figure 3 C-E). Excuse for us, but it is hard to use human embryonic stem cell lines for this study due to ethical issues in Japanese governmental regulations. The treatment of LY333531 and IWR-1-endo increased the expression of self-renewal marker genes/proteins and decreased the expression levels of SOX17, T, and PAX6 in these hiPSC lines in general. These results indicated that these chemical treatments in suspension culture were robust in general while addressing the observed variability among iPSC lines.

Additionally, there is a lack of information regarding the specific role of the two small molecules in these conditions.

In this revised manuscript, we have added data and discussion regarding the specific role of the two small molecules in these conditions in the Results and Discussion section. For using WNT signaling inhibitor, we hypothesized that adding Wnt signaling inhibitors may inhibit the spontaneous differentiation of hiPSCs into mesendoderm. Because exogenous Wnt signaling induces the differentiation of human pluripotent stem cells into mesendoderm lineages (Nakanishi *et al*, 2009; Sumi *et al*, 2008; Tran *et al*, 2009; Vijayaragavan *et al*, 2009; Woll *et al*, 2008). Also, endogenous expression and activation of Wnt signaling in pluripotent stem cells are involved in the regulation of mesendoderm differentiation potentials (Dziedzicka *et al*, 2021). For using PKC inhibitors, "To identify molecules with inhibitory activity on neuroectodermal differentiation, hiPSCs were treated with candidate molecules in suspension conditions. We selected these candidate molecules based on previous studies related to signaling pathways or epigenetic regulations in neuroectodermal development (reviewed in GiacomanLozano *et al*, 2022; Imaizumi & Okano, 2021; Sasai *et al*, 2021; Stern, 2024) or in pluripotency safeguards (reviewed in Hackett & Surani, 2014; Li & Belmonte, 2017; Takahashi & Yamanaka, 2016; Yagi *et al*, 2017)."

We also found that the expression of naïve pluripotency markers, *KLF2*, *KLF4*, *KLF5*, and *DPPA3*, were up-regulated in the suspension conditions treated with LY333531 and IWR-1-endo while the expression of *OCT4* and *NANOG* was at the same levels (Figure 5—figure supplement 2). Combined with RT-qPCR analysis data on 5 different hiPSC lines (Figure 3E), these results suggest that IWRLY conditions may drive hiPSCs in suspension conditions to shift toward naïve pluripotent states.

The authors have not attempted to elucidate the underlying mechanism other than RNA expression analysis.

Regarding the underlying mechanisms, we have added results and discussion in the revised manuscript. For Wnt activation in human pluripotent stem cells, several studies reported some WNT agonists were expressed in undifferentiated human pluripotent stem cells (Dziedzicka *et al.*, 2021; Jiang *et al*, 2013; Konze *et al*, 2014). In suspension culture, cell aggregation causes tight cell-cell interaction. The paracrine effect of WNT agonists in the cell aggregation may strongly affect neighbor cells to induce spontaneous differentiation into mesendodermal cells. Thus, we think that the inhibition of WNT signaling is effective to suppress the spontaneous differentiation into mesendodermal lineages in suspension culture.

For PKC beta activation in human pluripotent stem cells, we have shown that phosphorylated PKC beta protein expression is up-regulated in suspension culture than in adherent culture with western blotting (Figure 3 - figure supplement 1). The treatment of PKCβ inhibitor is effective to suppress spontaneous differentiation into neuroectodermal lineages. For future perspectives, it is interesting to examine (1) how and why PKCβ is activated (or phosphorylated), especially in suspension culture conditions, and (2) how and why PKCβ inhibition can suppress the neuroectodermal differentiation. Conversely, it is also interesting to examine how and why PKCβ activation is related to neuroectodermal differentiation.

For these reasons some aspects of the manuscript need to be extended:(1) It is crucial for authors to specify the culture media used for suspension cultures. In the Materials and Methods section, the authors mentioned that cells in suspension were cultured in either StemFit AK02N medium, 415 StemFit AK03N (Cat# AK03N, Ajinomoto, Co., Ltd., Tokyo, Japan), or StemScale PSC416 suspension medium (A4965001, Thermo Fisher Scientific, MA, USA). The authors should clarify in the text which medium was used for suspension cultures and whether they observed any differences among these media.

Sorry for this confusion. Basically in this study, we use StemFit AK02N medium (Figure 1-5, 7-9). For bioreactor experiments (Figure 6), we use StemFit AK03N medium, which is free of human and animalderived components and GMP grade. To confirm the effect of IWRLY chemical treatment, we use StemScale suspension medium (Figure 4 - figure supplement 1) and mTeSR1 medium (Figure 4 - figure supplement 2 and Figure 8 - figure supplement 1). In the revised manuscript we clarified which medium was used for suspension cultures in the Results and Materials and Methods section.

Although we have not compared directly among these media in suspension culture (, which is primarily out of the focus of this study), we have observed some differences in maintaining self-renewal characteristics, preventing spontaneous differentiation (including tendencies to differentiate into specific lineages), stability or variation among different experimental times in suspension culture conditions. Overcoming these heterogeneity caused by different media, the IWRLY chemical treatment stably maintain hiPSC self-renewal in general. We have added this issue in the Discussion section.

(2) In the Materials and Methods section, the authors mentioned that they used multiple cell lines for this study. However, it is not clear in the text which cell lines were used for various experiments. Since there is considerable variation among iPSC lines, I suggest that the authors simultaneously compare 2 to 3 pluripotent stem cell lines for expansion, differentiation, etc.

Thank you for this careful suggestion. We have added more results on the simultaneous comparison using StemFit AK02N medium in 5 different hiPSC lines (Figure 3 C-E) and using mTeSR1 medium in 4 different hiPSC lines (Figure 4 - figure supplement 2). From both results, we have shown that the treatment of LY333531 and IWR-1-endo was beneficial in maintaining the self-renewal of hiPSCs while suppressing spontaneous differentiation.

(3) Single-cell sorting can be confusing. Can iPSCs grown in suspensions be single-cell sorted?

Additionally, what was the cloning efficiency? The cloning efficiency should be compared with adherent cultures.

Sorry for this confusion. With our method, iPSCs grown in IWRLY suspension conditions can be singlecell sorted. We have improved the clarity of the schematics (Figure 7A). Also, we added the data on the cloning efficiency, which are compared with adherent cultures (Figure 7B). The cloning efficiency of adherent cultures was around 30%. While the cloning efficiency of suspension cultures without any chemical treatment was less than 10%, the IWR-1-endo treatment in the suspension cultures increased the efficiency was more than 20%. However, the treatment of LY333531 decreased the efficiency. These results indicated that the IWR-1-endo treatment is beneficial in single-cell cloning in suspension culture.

(4) The authors have not addressed the naïve pluripotent state in their suspension cultures, even though PKC inhibition has been shown to drive cells toward this state. I suggest the authors measure the expression of a few naïve pluripotent state markers and compare them with adherent cultures

Thank you for this insightful comment. In the revised manuscript, we have added the data of RT-qPCR in 5 different hiPSC lines and specific gene expression from RNA-seq on naïve pluripotent state markers (Figure 3E and Figure 5 - figure supplement 2), respectively. Interestingly, the expression of *KLF2, KLF4, KLF5, and DPPA3* is significantly up-regulated in IWRLY conditions. These results suggested that IWRLY suspension conditions drove hiPSCs toward naïve pluripotent state.

**Recommendations for the authors:**

**Reviewer #1 (Recommendations For The Authors):**
Overall, I feel that this study is very interesting and comprehensive, but has significant weaknesses in the bioprocessing aspects. More optimization data is required for the suspension culture to truly show that the differentiation they are observing is not an artifact of a non-optimized protocol.

Thank you for your thoughtful comments. Following your comments, we have performed several experiments to optimize the bioreactor conditions in revised manuscripts. We tested several cell seeding densities and several stirring speeds with or without WNT/PKCβ inhibitors (Figure 6—figure supplement 1). From these optimization experiments, we found that 1 - 2 x 105 cells/mL of the seeding densities and 50 - 150 rpm of the stirring speeds were applicable in the proliferation of these cells. Also, PKCβ and Wnt inhibitors suppressed spontaneous differentiation in bioreactor conditions regardless with acceptable stirring speeds. As for the impeller shape and reactor design, we just used commonly-used ABLE's bioreactor for 30 mL scale and Eppendorf's bioreactors for 320 mL scale, which had been designed and used for human pluripotent stem cell culture conditions in previous studies, respectively Matsumoto et al., 2022 (doi: 10.3390/bioengineering9110613); Kropp et al., 2016 (doi:10.5966/sctm.2015-0253). We cited these previous studies in the Results section. We believe that these additional data and explanation are sufficient to satisfy your concerns on the optimization of bioreactor experiments.

**Reviewer #2 (Recommendations For The Authors):**
The following comments should be addressed by the authors to improve the manuscript:(1) Abstract: '...a scalable culture system that can precisely control the cell status for hiPSCs is not developed yet.' There were previous reports for a scalable iPSC culture system so I would suggest toning down/rephrasing this point: eg that improvement in a scalable iPSC culture system is needed.

Thank you for this careful suggestion. Following this suggestion, We have changed the sentence as "the improvement in a scalable culture system that can precisely control the cell status for hiPSCs is needed."

(2) Line 71: please specify what media was used as a 'conventional medium' for suspension culture, was it Stemscale?

As suggested, we specified the media as StemFit AK02N used for this experiment.

(3) Fig 1E: It's not easy to see gating in the FACS plots as the threshold line is very faint, please fix this issue.

As suggested, we used thicker lines for the gating in the FACS plots (Figure 1E).

(4) Fig 1G-J, Fig 2D-H: The RNAseq figures appeared pixelated and the resolution of these figures should be improved. The x-axis label for Fig 1H is missing.

We have improved these figures in their resolution and clarity. Also, we have added the x-axis label as "enrichment distribution" for gene set enrichment analysis (GSEA) in Figures 1H, 5F, and 5- figure supplement 1B.

(5) Line 103-107: 'Since Wnt signaling induces the differentiation of human pluripotent stem cells into mesendoderm lineages, and is endogenously involved in the regulation of mesendoderm differentiation of pluripotent stem cells.....'. The two points seem the same and should be clarified.

Sorry for this unclear description. We have changed this description as "Exogenous Wnt signaling induces the differentiation of human pluripotent stem cells into mesendoderm lineages (Nakanishi *et al*, 2009; Sumi *et al*, 2008; Tran *et al*, 2009; Vijayaragavan *et al*, 2009; Woll *et al*, 2008). Also, endogenous expression and activation of WNT signaling in pluripotent stem cells are involved in the regulation of mesendoderm differentiation potentials (Dziedzicka *et al*, 2021; Jiang *et al*, 2013)." With this description, we hope that you will understand the difference of two points.

(6) Line 113: 'In samples treated with inhibitors' should be 'In samples treated with Wnt inhibitors'.

Thank you for this careful suggestion. We have corrected this.

(7) Line 115: '....there was no reduction in PAX6 expression.' That's not entirely correct, there was a reduction in PAX6 in IWR-1 endo treatment compared to control suspension culture (is this significant?), but not consistently for IWP-2 treatment. Please rephrase to more accurately describe the results.

Sorry for this inaccurate description. We have corrected this phrase as "there was only a small reduction in PAX6 expression in the IWR-1-endo-treated condition and no reduction in the IWP2-treated condition" as recommended.

(8) It's critical to show that the effect of the suspension culture system developed here can maintain an undifferentiated state for multiple hiPSC lines. I think the author did test this in multiple cell lines, but the results are scattered and not easy to extract. I would recommend adding info for the hiPSC line used for the results in the legend, eg WTC11 line was used for Figure 3, 201B7 line was used for Figure 2. I would suggest compiling a figure that confirms the developed suspension system (IWR-1 +LY) can support the maintenance of multiple hiPSC lines.

Thank you for this insightful suggestion. We have added data on hiPSC maintenance across 5 hiPSC lines in suspension culture using StemFit AK02N medium simultaneously (Figure 3C - E) and on hiPSC maintenance across 4 hiPSC lines in suspension culture using mTeSR1 medium simultaneously (Figure 4 - figure supplement 2). Together, the treatment of LY333531 and IWR-1-endo in these media reversed the decreased expression of these undifferentiated markers and suppressed the increased expression of differentiation markers in suspension culture conditions. These results show that these chemical treatment produced a consistent robust effect in hiPSC maintenance across multiple cell lines.

(9) Line 166: Please use the correct gene nomenclature format for a human gene (italicised uppercase) throughout the manuscript. Also, list the full gene name rather than PAX2,3,5.

Sorry for the incorrectness of the gene names. We have corrected them.

(10) Please improve the resolution for Figure 4D.

We have provided clearer images of Figure 4D.

(11) In the first part of the study, the control condition was referred to as 'suspension culture' with spontaneous differentiation, but in the later parts sometimes the term 'suspension culture' was used to describe the IWR1+LY condition (ie lines 271-272). I would suggest the authors carefully go through the manuscript to avoid misinterpretation on this issue.

Thank you for this careful suggestion. To avoid this misinterpretation on this issue, we use 'suspension culture' for just the conventional culture medium and 'LYIWR suspension culture' for the culture medium supplemented with LY333531 and IWR1-endo in this manuscript.

(12) Figure 5: It is impressive to demonstrate that the IWR1+LY suspension culture enables large-scale expansion of a clinical-grade hiPSC line using a bioreactor, yielding 300 vials/passage. Can the author add some information regarding cell yield using a conventional adherent culture system in this cell line? This will provide a comparison of the performance of the IWR1+LY suspension culture system to the conventional method.

Thank you for this valuable suggestion. We have provided information regarding cell yield using a conventional adherent culture system in this cell line in the Results as "Since the population doubling time (PDT) of this hiPSC line in adherent culture conditions is 21.8 - 32.9 hours at its production (https://www.cira-foundation.or.jp/e/assets/file/provision-of-ips-cells/QHJI14s04_en.pdf), this proliferation rate in this large scale suspension culture is comparable to adherent culture conditions."

(13) Line 273: For testing the feasibility of using IWR1+LY media to support the freeze and thaw process, the author described the cell number and TRA160+/OCT4+ cell %. How is this compared to conventional media (eg E8)? It would be nice to see a head-to-head comparison with conventional media, quantification of cell count or survival would be helpful to determine this.

For this issue, we attempted a direct freeze and thaw process using conventional media, StemFit AK02N in 201B7 line (Figure 8) or mTeSR1 in 4 different hiPSC lines(Figure 8 - figure supplement 1) with or without IWR1+LY. However, since the hiPSCs cultured in suspension culture conditions without IWR1+LY quickly lost their self-renewal ability, these frozen cells could not be recovered in these conditions nor counted. Our results indicate that the addition to IWR1+LY in the thawing process support the successful recovery in suspension conditions.

(14) More details of the passaging method should be added in the method section. Do you do cell count following accutase dissociation and replate a defined density (eg 1x10^5/ml)?

Yes. We counted the cells in every passage in suspension culture conditions. We have added more explanation in the Materials and Methods as below.

"The dissociated cells were counted with an automatic cell counter (Model R1, Olympus) with Trypan Blue staining to detect live/dead cells. The cell-containing medium was spun down at 200 rpm for 3 minutes, and the supernatant was aspirated. The cell pellet was re-suspended with a new culture medium at an appropriate cell concentration and used for the next suspension culture."

(15) The IWR1+LY suspension culture system requires passage every 3-5 days. Is there still spontaneous differentiation if the hiPSC aggregate grows too big?

Thank you for this insightful question.

Yes. The size of hiPSC aggregates is critical in maintaining self-renewal in our method as previous studies showed. Stirring speed is a key to make the proper size of hiPSC aggregates in suspension culture. Also, the culture period between passages is another key not to exceed the proper size of hiPSC aggregates. Thus, we keep stirring speed at 90 rpm (135 rpm for bioreactor conditions) basically and passaging every 3 - 5 days in suspension culture conditions.

(16) Several previous studies have described the development of hiPSC suspension culture system using hydrogel encapsulation to provide biophysical modulation (reviewed in PMID: 32117992). In comparison, it seems that the IWR1+LY suspension system described here does not require ECM addition which further simplifies the culture system for iPSC. It would be good to add more discussion on this topic in the manuscript, such as the potential role of the E-cadherin in mediating this effect - as RNAseq results indicated that CDH1 was upregulated in the IWR1+LY condition.

Thank you for this valuable suggestion. We have added more discussion on this topic in the Discussion section as below.

"Thus, our findings show that suspension culture conditions with Wnt and PKCβ inhibitors (IWRLY suspension conditions) can precisely control cell conditions and are comparable to conventional adhesion cultures regarding cellular function and proliferation. Many previous 3D culture methods intended for mass expansion used hydrogel-based encapsulation or microcarrier-based methods to provide scaffolds and biophysical modulation (Chan *et al*, 2020). These methods are useful in that they enable mass culture while maintaining scaffold dependence. However, the need for special materials and equipment and the labor and cost involved are concerns toward industrial mass culture. On the other hand, our IWRLY suspension conditions do not require special materials such as hydrogels, microcarriers, or dialysis bags, and have the advantage that common bioreactors can be used. "

"On the other hand, it is interesting to see whether and how the properties of hiPSCs cultured in IWRLY suspension culture conditions are altered from the adherent conditions. Our transcriptome results in comparison to adherent conditions show that gene expression associated with cell-to-cell attachment, including E-cadherin (CDH1), is more activated. This may be due to the status that these hiPSCs are more dependent on cell-to-cell adhesion where there is no exogenous cell-to-substrate attachment in the three-dimensional culture. Previous studies have shown that cell-to-cell adhesion by E-cadherin positively regulates the survival, proliferation, and self-renewal of human pluripotent stem cells (Aban *et al*, 2021; Li *et al*, 2012; Ohgushi *et al*, 2010). Furthermore, studies have shown that human pluripotent stem cells can be cultured using an artificial substrate consisting of recombinant E-cadherin protein alone without any ECM proteins (Nagaoka *et al*, 2010). Also, cell-to-cell adhesion through gap junctions regulates the survival and proliferation of human pluripotent stem cells (Wong *et al*, 2006; Wong *et al*, 2004). These findings raise the possibility that the cell-to-cell adhesion, such as E-cadherin and gap junctions, are compensatory activated and support hiPSC self-renewal in situations where there are no exogenous ECM components and its downstream integrin and focal adhesion signals are not forcedly activated in suspension culture conditions. It will be interesting to elucidate these molecular mechanisms related to E-cadherin in the hiPSC survival and self-renewal in IWRLY suspension conditions in the future."

**Reviewer #3 (Recommendations For The Authors):**
(1) I am a bit confused about the passage of adherent cultures. The authors claim that they used EDTA for passaging and plated cells at a density of 2500 cells/cm2. My understanding is that EDTA is typically used for clump passaging rather than single-cell passaging.

Sorry about this confusion. We routinely use an automatic cell counter (model R1, Olympus) which can even count small clumpy cells accurately. Thus, we show the cell numbers in the passaging of adherent hiPSCs.

(2) Figure 2D- The authors have not directly compared IWR-1-endo with IWR-1-endo+Go6983 for the expression of T and SOX17, a simultaneous comparison would be an interesting data.

As recommended, we have added the data that directly compared IWR-1-endo with IWR-1endo+Go6983 for the expression of *T* and *SOX17* in Figure 2D. The addition of IWR-1-endo alone decreased the expression of *T* and *SOX17*, but not *PAX6*, which were similar to the data in Figure 2C.

(3) Oxygen levels play a crucial role in pluripotency maintenance. Could the authors please specify the oxygen levels used for culturing cells in suspension?

Sorry for not mentioning about oxygen levels in this study. We basically use normal oxygen levels (i.e., 21% O2) in suspension culture conditions. We have explained this in the Materials and Methods section.

(4) Figure supplement 1 (G and H): In the images, it is difficult to determine whether the green (PAX6 and SOX17) overlaps with tdT tomato. For better visualization, I suggest that the authors provide separate images for the green and red colors, as well as an overlay.

Sorry for these unclear images. We have provided separate images for the green and red colors, as well as an overlay in Figure 1- figure supplement 1 G and H.

(5) The authors have only compared quantitatively the expression of TRA-1-60 for most of the figures. I suggest that the authors quantitatively measure the expression of other markers of undifferentiated stem cells, such as NANOG, OCT4, SSEA4, TRA-1-81, etc.

We have added the quantitative data of the expression of markers of undifferentiated hiPSCs including NANOG, OCT4, SSEA4, and TRA-1-60 on 5 different hiPSC lines in Figure 3 C-E.

(6) In Figure 2D, the authors have tested various small molecules but the rationale behind testing those molecules is missing in the text.

These molecules are chosen as putatively affecting neuroectodermal induction from the pluripotent state.

We have added the rationale with appropriate references in the Results section as below.

"We have chosen these candidate molecules based on previous studies related to signaling pathways or epigenetic regulations in neuroectodermal development (reviewed in Giacoman-Lozano *et al*, 2022; Imaizumi & Okano, 2021; Sasai *et al*, 2021; Stern, 2024) or in pluripotency safeguards (reviewed in Hackett & Surani, 2014; Li & Belmonte, 2017; Takahashi & Yamanaka, 2016; Yagi *et al*, 2017) (Figure 2A; listed in Supplementary Table 1). "

(7) In the beginning authors used Go6983 but later they switched to LY333531, the reasoning behind the switch is not explained well.

To explain the reasons for switching to LY333531 from Go6983 clearly, we reorganized the order of results and figures. In short, we found that the suppression of PAX6 expression in hiPSCs cultured in suspension conditions was observed with many PKC inhibitors, all of which possessed PKCβ inhibition activity (Figure 2—figure supplement 2B-D). Also, elevated expression of PKCβ in suspension-cultured hiPSCs could affect the spontaneous differentiation (Figure 3—figure supplement 1A-C). To further explore the possibility that the inhibition of PKCβ is critical for the maintenance of self-renewal of hiPSCs in the suspension culture, we evaluated the effect of LY333531, a PKCβ specific inhibitor. The maintenance of suspension-cultured hiPSCs is specifically facilitated by the combination of PKCβ and Wnt signaling inhibition (Figure 3A and B; Figure 2—figure supplement 1). Last, we performed longterm culture for 10 passages in suspension conditions and compared hiPSC growth in the presence of LY333531 or Go6983. LY333531 was superior in the proliferation rate and maintaining OCT4 protein expression in the long-term culture (Figure 4). Thus, we used IWR-1-endo and LY333531 for the rest of this study.

(8) I suggest the authors measure cell death after the treatment with LY+IWR-1-endo.

Thank you for this valuable suggestion. We have measured cell death after the treatment with LY+IWR1-endo and found that the chemical combination had no or little effects on the cell death. We have added data in Figure 3—figure supplement 2 and the description in the Results section as below. "We also examined whether the combination of PKCb and Wnt signaling inhibition affects the cell survival in suspension conditions. In this experiment, we used another PKC inhibitor, Staurosporine (Omura *et al*, 1977), which has a strong cytotoxic effect as a positive control of cell death in suspension conditions. The addition of IWR-1-endo and LY333531 for 10 days had no effects on the apoptosis while the addition of Staurosporine for 2 hours induced Annexin-V-positive apoptotic cells (Figure 3—figure supplement 2). These results indicate that the combination of PKCb and Wnt signaling inhibition has no or little effects on the cell survival in suspension conditions."

(9) The authors have performed reprogramming using episomal vectors and using Sendai viruses. In both the protocols authors have added small molecules at different time points, for episomal vector protocol at day 3 and Sendai virus protocol at day 23. Why is this different?

Thank you for this insightful question. We intended that these differences should be reflected in the degree of the expression from these reprogramming vectors. The expression of reprogramming factors from these vectors should suppress the spontaneous differentiation in reprogramming cells. Sendai viral vectors should last longer than episomal plasmid vectors. Thus, we thought that adding these chemical inhibitors for episomal plasmid vector conditions from the early phase of reprogramming and for Sendai viral vector conditions from the late phase of reprogramming. For future perspectives, we might further need to optimize the timing of adding these molecules.

(10) The protocol for three germ layer differentiation using a specific differentiation medium requires further elaboration. For instance, the authors mentioned that suspension cultures were transferred to differentiation media but did not emphasize the cell number and culture conditions before moving the cultures to the differentiation media.

Sorry for this unclear description. We have added the explanation on the cell number and culture conditions before moving the cultures to the differentiation media in the Materials and Methods section as below.

"As in the maintenance conditions, 4 × 105 hiPSC were seeded in one well of a low-attachment 6-well plate with 4 mL of StemFit AK02N medium supplemented with 10 µM Y-27632. This plate was placed onto the plate shaker in the CO2 incubator. Next day, the medium was changed to the germ layer specific differentiation medium."